
# Interfacially adsorbed bubbles determine the shape of droplets

Alessio Squarcini[1][*] and Antonio Tinti[2][†]

**1** Institut für Theoretische Physik, Universität Innsbruck,
Technikerstrasse 21A, A-6020 Innsbruck, Austria
**2** Dipartimento di Ingegneria Meccanica e Aerospaziale,
Sapienza Università di Roma, via Eudossiana 18, 00184 Rome, Italy

[*] alessio.squarcini@uibk.ac.at ,  [†] antonio.tinti@uniroma1.it

## Abstract

The characterization of density correlations in the presence of strongly fluctuating interfaces has always been considered a difficult problem in statistical mechanics. Here we study – by using recently developed exact field-theoretical techniques – density correlations for an interface with endpoints on a wall forming a droplet in 2D. Our framework applies to interfaces entropically repelled by a hard wall as well as to wetting transitions. In the former case bubbles adsorbed on the interface are taken into account by the theory which yields a systematic treatment of finite-size corrections to one- and two-point functions and show how these are related to Brownian excursions. Our analytical predictions are confirmed by Monte Carlo simulations without free parameters. We also determine one- and two-point functions at wetting by using integrable boundary field theory. We show that correlations are long ranged for entropic repulsion and at wetting. For both regimes we investigate correlations in momentum space by generalizing the notion of interface structure factor to semi-confined systems. Distinctive signatures of the two regimes manifest in the structure factor through a term that we identify on top of the capillary-wave one.



# 1  Introduction

The study of interfacial phenomena in the presence of boundaries constitutes one of the most important chapters in statistical mechanics in which theory faces with experiments and technological applications [1]. The archetypical scenario is the one of a liquid on a substrate in coexistence with its vapour. In general, a liquid layer intrudes between the substrate and the vapour. The transition from a microscopic (finite) to a macroscopic (infinite) layer thickness as the temperature is increased towards a wetting temperature $T_w$ is termed *wetting transition* [2,3]. Beside the morphological change of the interfacial shape, the latter is a *surface phase transition* accompanied by the divergence of thermodynamic quantities whose singular behavior is characterized in terms of critical exponents and surface universality classes [1,4–8]. From the theoretical side, boundary-induced effects on near-critical systems have been intensively studied by means of several techniques ranging from mean field theory,[1] perturbative field theory [9–11], and numerical simulations [12]. Despite the vast literature, exact results in the field of interfacial phenomena are scarce and predominantly limited to the two dimensional (2D) Ising model [13]. This circumstance naturally raises the legitimate question whether the already existing exact findings are related to the solvability of the lattice model, or whether they are universal, in the sense that they are shared by other universality classes. For more than 40 years the Ising model in 2D allowed for the investigation of interfacial phenomena with exact techniques [13]. In a seminal paper, D. B. Abraham [14] provides the first example of an exactly solvable system exhibiting a wetting transition. Almost half a century ago Wertheim [15] predicted – by using integral equation theories for the liquid state [16] – that density fluctuation at the interface separating coexisting phases are long-ranged and confined within the interfacial region. Such findings influenced the development of the field of interfacial phenomena but at the same time they also posed questions that so far have received partial answers. In particular, how to compute correlation functions within the interfacial region in an exact fashion by going beyond the Ising model in 2D? How to test Wertheim's prediction in the strongly fluctuating regime of 2D systems? Does a wetting transition affect the long-range character of interfacial fluctuations? If so, is it possible to measure them via the interface

---

[1]See, e.g., Sec. C of Ref. [9].

structure factor? How to quantify finite-size corrections due to the finiteness of the interface extent? In the present work we provide answers to these questions, a task that now is possible thanks to the exact field theory of phase separation developed in the last decade [17–22].

Interfaces arising from phase separation in 2D are particularly intriguing for a twofold reason. On the one side because thermal fluctuations have strong effects on correlations and effective descriptions based on mean-field theories are – *de facto* – unable to capture the exact form of correlations. On the other hand, the 2D case can be analyzed in a mathematically precise and controllable fashion by exploiting analytical techniques yielding exact solutions. The intrinsic inadequacy of mean-field theory treatment below the upper critical dimension is nicely illustrated by the case of the 2D Ising model in bulk (i.e., the unbounded plane). The celebrated Onsager's solution yields the exact result for the decay of scaled truncated two-point function, $g_2(r) \sim r^{-2} \exp(-2r)$ [23]. This result is in sharp contrast with the prediction of Ornstein-Zernike theory, $g_2^{(\mathrm{OZ})}(r) \sim r^{-1/2} \exp(-r)$, which exhibits an anomalous power-law exponent [16, 24], a phenomenon that goes under the name of *Kadanoff-Wu anomaly* [25]. While the exact form of two-point correlation functions for the Ising model in the unbounded plane is nowadays textbook knowledge [26], the exact analytic form of correlations in the presence of *strongly fluctuating interfaces*[2] is largely unknown to date up to some notable exceptions, this because the actual shape of boundaries, the implementation of boundary conditions, and features of the universality class, come into play altogether and the way they interplay is far from being trivial.

Although the density profile for the interface of the Ising model on the strip has been known since a pioneering work of D. B. Abraham in 1974 [27], the two-point function for other universality classes in such a geometry has been obtained 40 years later [22]. The next problem in this hierarchy is the droplet on the half-plane shown in Fig. 1 (*a*). For this geometry we know the density profile for the Ising model [14] while the exact calculation of the two-point function resisted to analytical investigations until recent times [28].

The analytical progresses mentioned above have been put forward in the last decade with the exact theory of phase separation in 2D for a broad range of universality classes [17–22]. By exploiting low-energy properties of field theories it has been possible to calculate one- and two-point correlation functions from the underlying field theory in the presence of an interface on a finite strip [22]. More recently, many-body correlation functions have been computed analytically [29] and the predictions have been successfully confirmed by accurate Monte Carlo simulations (MC) simulations in the absence of free parameters [30–32].

Besides exactly solvable models, the typical approach to the study of interfacial correlations – especially for three dimensional systems – follows the route of interfacial Hamiltonian models and capillary wave theory [33] (see [34] and references therein). In such phenomenological theories it is assumed that interfacial undulations are Gaussian [35,36] and correlations are investigated in both real and momentum space, in the latter case through the notion of *interface structure factor* (ISF) [37–41]. However, the *exact* form of correlations both in real and momentum space for the case in which the wall undergoes a wetting transition is still missing.

In the present work we fill this gap by showing how the first-principle approach based on recently developed field-theoretical techniques [20, 22] yields exact results in 2*D* for density profiles and interfacial correlations for the droplet of Fig. 1 both in real and momentum space. The formalism is applied to two interesting regimes: i) a pinned interface experiencing entropic repulsion by a hard wall, ii) temperatures close to the wetting transition in which the interface binds/unbinds from the wall and leads to a diverging wetting layer. In the regime

---

[2]With this terminology we refer to interfaces pinned at boundaries, which are inevitably non uniform in boundary conditions. See Chap. 21 of Ref. [26] for results on correlation functions in the presence of a uniform boundary in the 2D Ising model within the framework of integrable boundary quantum field theory.

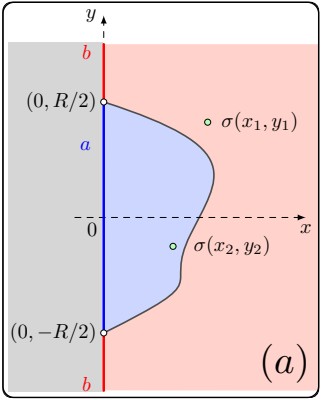
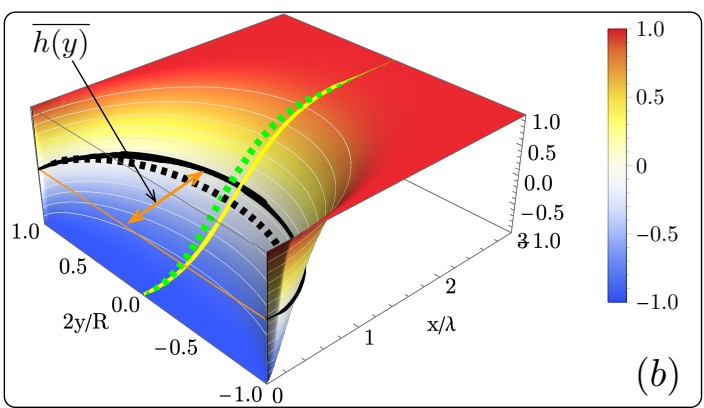

Figure 1: Panel (a): the half-plane with boundary conditions leading to the formation of a droplet with pinning points in $(0, \pm R/2)$. Panel (b): the order parameter profile as a function of $x$ and $y$ rescaled to the bulk value. The black curve is an arc of ellipse corresponding to the isoline at which the density attains the zero value, the latter defines a sharp interface with average distance from the wall $\overline{h(y)} \propto \sqrt{R - (2y)^2/R}$ given by the mean value of the Brownian excursion (28). The dashed black line is the isoline at vanishing value including the finite-size correction $\propto \mathscr{A}$ in (12). The yellow line is the density profile at leading order sliced along the $x$-axis; see (13). The dashed green line is the density profile including the interface structure effects for $R/\xi_{\mathrm{b}} = 20$; see (12).

of entropic repulsion we show that interface structure effects, missing in previous treatments, cannot be ignored and their effect resolve a longstanding discrepancy observed in early simulations [42].

The picture that emerges is that both regimes of entropic repulsion and wetting are characterized by long-range correlations along the interface but these correlations are confined within the interfacial region. This can be concluded by inspection of the density-density correlation function in real space, something that we find in closed-form. To further establish a connection with observables considered in experiments [43–45], we extend the notion of ISF to semi-confined systems and analyze correlations in momentum space. In particular, we show that for large wavenumber $q$ both regimes share the typical form of the ISF predicted by capillary wave theory, i.e., $S(q) \sim A/q^2$. The coefficient is $A = (\rho_l - \rho_v)^2/\gamma_{lv}$ where $\gamma_{lv}$ is the surface tension of the liquid-vapour interface, $\rho_l$ and $\rho_v$ are the bulk densities in the liquid and vapour phases, respectively. The $1/q^2$ behavior emerges for wave-numbers $q$ much bigger than a lower momentum cutoff $\Lambda$ that is regime-dependent. For the case of entropic repulsion $\Lambda^{(\mathrm{er})} = 1/R$ is the inverse of the sessile droplet extent, the latter plays the role of the capillary length mimicking by the pinning. For the case of wetting instead $\Lambda^{(\mathrm{w})} = 1/\xi_{\parallel}$ and $\xi_{\parallel} \sim (T_{\mathrm{w}} - T)^{-\nu_{\parallel}}$ dictates the typical length of interfacial fluctuations along the interface. In capillary-wave theory the large-distance behavior yielding the small momentum singularity $1/q^2$ is the signature of a Goldstone mode related to a continuous symmetry in the system [46]. This symmetry is translational invariance, which asymptotically emerges at the wetting transition since the interface can be displaced with no cost in energy. Surprisingly, the $1/q^2$ behavior emerges also in a system with finite but large extent in which translational invariance can be restored only within a limiting procedure in which $R \to \infty$; this is the case of a sessile droplet of diameter $R$ experiencing entropic repulsion in two dimensions. Although wetting and entropic repulsion are characterized by the same behavior of $S(q)$ for $q \gg \Lambda$, characteristic signatures distinctive of the two regimes appear in the form of corrections that involve higher powers of the wave number.

This paper is organized as follows: Sec. 2 illustrates in a rather pedagogical manner the key ideas underlying the exact field theory of phase separation and its connection with the language of interfacial phenomena. In Sec. 3 we briefly motivate the choice of the model used in simulations and the observables of interest. Sec. 4 presents the comparison between theory and MC simulations for the regime of entropic repulsion in the Ising model. In Sec. 5 we recall the essential phenomenological features of a wetting transition and its connection with exact results in field theory. We then illustrate our theoretical predictions for one- and two-point correlation functions of the order parameter and discuss them. We then proceed with the analysis of correlations in momentum space in Sec. 6. Our conclusions and perspectives for future research are presented in Sec. 7. Details about the field-theoretic calculations of one-point functions at wetting are supplied in Appendix A together with a derivation from first principles of Antonov's rule for wetting. Two-point functions of the energy density at wetting are considered in Appendix B. The calculation is sketched and the full details of it, including also the calculation of order parameter correlation functions will be disclosed in a forthcoming publication [89]. Appendix C contains the details for the material covered in Sec. 6.

## 2 Theoretical framework in a nutshell

The aim of this section is to summarize in a succinct manner the key ideas underlying the exact field theory of phase separation developed in [17–20]. Quite generally, it has been possible to show that certain notions in field theory admit a one-to-one correspondence with the theory of interfacial phenomena [1, 5]. The two languages are brought in touch by means of the dictionary summarized in Tab. 1 whose explanation is provided in the following.

We consider the scaling limit of a ferromagnetic spin model below the critical temperature $T_c$ but sufficiently close to it in order to use a continuum description. The near-critical behavior in 2D can be described by analytic continuation of a $(1 + 1)$-relativistic quantum field theory to a $2D$ Euclidean field theory in the plane ($y = -it$). Homogeneous fluid phases in thermodynamic equilibrium can be described as the set of degenerate ground states in field theory, which we denote as $\{|0_a\rangle\}_{a=1,\dots,n}$. In a system exhibiting coexistence of two phases, say $|0_a\rangle$ and $|0_b\rangle$, an interface separates two clusters occupied by the distinct phases, as shown in Fig. 2. This is the archetypical situation of the equilibrium of two phases like a liquid in coexistence with its vapour phase. The interface separating phases $a$ and $b$ corresponds, in field theory, to the trajectory in the $(1+1)$ dimensional space time $(x, t)$ of a particle excitation connecting the vacuum state $|0_a\rangle$ to $|0_b\rangle$, as depicted in Fig. 2. Excitations in 2D have topological nature and are termed kinks. The corresponding relativistic field theory is formulated in terms of massive particles states denoted $|K_{ab}(\theta)\rangle$ with energy-momentum

$$P^\mu = (e, p) = (m \cosh \theta, m \sinh \theta). \tag{1}$$

$m$ is the kink mass, $\theta$ is the rapidity which trivially ensures the energy-momentum condition $P^\mu P_\mu = e^2 - p^2 = m^2$, $\mu = 0, 1$. Indexes are raised and lowered with the metric tensor in Minkowski space time given by the Pauli matrix $g^{\mu\nu} = \text{diag}(1, -1) = \sigma^z$.

The equivalent of the number density field in liquid state theories is represented by the order parameter field, or spin, $\sigma(x, y)$ in the field-theoretical language. For a uniform system filled by a pure phase $|0_a\rangle$ the average density is translationally invariant and given by the vacuum expectation value $\langle \sigma \rangle_a \sim (T_c - T)^\beta$ in the near-critical region. For a system exhibiting phase separation, like the droplet we are examining in this paper, the average density field is a function that interpolates between $\langle \sigma \rangle_a$ and $\langle \sigma \rangle_b$. In general, the calculation of the density profile reduces to the analysis of a matrix element between the spin field $\sigma$ and the asymptotic one-particle kink states. Leaving aside the involved technicalities, field theory shows that the

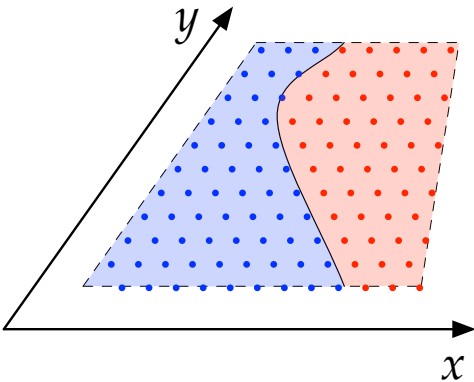

Figure 2: The worldline of a particle moving in the $(1+1)$-dimensional space time corresponds to the interface separating the left and right clusters.

derivative of the leading order density profile along the $x$ direction is proportional to the probability density of a Brownian walker emitted in the lower pinning point and absorbed in the upper pinning point. This result is encoded in the exact expressions

$$\partial_x \langle \sigma(x,y) \rangle_{ab} = (\langle \sigma \rangle_a - \langle \sigma \rangle_b) P(x,y), \tag{2}$$

and for the energy-density field

$$\langle \varepsilon(x,y) \rangle_{ab}^{c} \propto P(x,y). \tag{3}$$

On the strip, $P(x,y)$ is the probability density of a Brownian bridge, while the half-plane geometry $P(x,y)$ is the probability density of the so-called Brownian excursion;[3] the latter is a Brownian bridge constrained to the region $x > 0$. It is worth emphasizing that interfacial models typically assume a priori (2) with a Gaussian form for $P(x,y)$. The field-theory viewpoint is logically different: the left hand side of (2) is determined from first principles and it is found that (2) is indeed satisfied when $P(x,y)$ is the Brownian excursion.

Field theory then allows to reconstruct the density profile by averaging over configurations in which the interface is a sharp entity separating the bulk densities $\langle \sigma \rangle_a$ and $\langle \sigma \rangle_b$. By fixing the correct asymptotic densities it follows that the leading order form of the density profile is

$$\langle \sigma(x,y) \rangle_{ab} = \int_0^\infty du\, P(u,y) \sigma(x|u), \tag{4}$$

with the sharp profile $\sigma(x|u) = \langle \sigma \rangle_a$ if $x < u$ and $\sigma(x|u) = \langle \sigma \rangle_b$ if $x > u$, where $u$ is the point at which the sharp interface crosses the horizontal axis $y = 0$. As a result, the density profile is basically the cumulative distribution function of the Brownian excursion.

Field theory allows for an exact treatment of finite size correction, which in general can be systematized in the form of an expansion in powers of $R^{-1/2}$. The interface structure correction at order $R^{-1/2}$ still admits a probabilistic interpretation provided the sharp profile is endowed with a correction term localized on the interface. The resulting conditional magnetization has the form

$$\sigma(x|u) = \langle \sigma \rangle_a \theta(u-x) + \langle \sigma \rangle_b \theta(x-u) + \mathscr{A} \delta(x-u), \tag{5}$$

up to additional terms that yield corrections at order $R^{-1}$ and higher. Although this may appear a guess based on phenomenological considerations, it is consistent with exact field theoretical

---

[3]The emergence of random walk descriptions in interfacial phenomena is further discussed in Sec. 4.2.

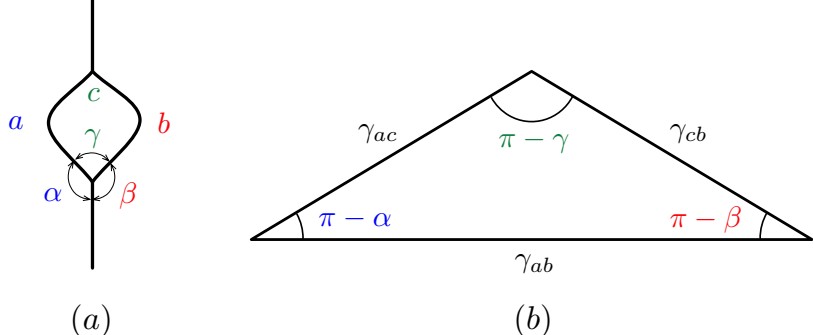

Figure 3: Panel (a): a bubble of phase $c$ adsorbed along the $a-b$ interface forming a contact angle $\gamma$. Panel (b) the Neumann triangle corresponding to the three phases $a$, $b$, and $c$ meeting at a common point with angles $\alpha$, $\beta$, and $\gamma$.

results, both on the strip [17] and on the half-plane [28]. Moreover, field theory allows us to fix the exact value of the amplitude $\mathscr{A}$, which depends on both the universality class of the model and the geometry. For the Ising model on the strip $\mathscr{A}_{\text{Ising}}^{(\text{strip})} = 0$ while for the $q$-state Potts model on the strip the amplitude takes a non-vanishing value [17].

Quite interestingly, a non-vanishing $\mathscr{A}$ on the strip is associated to those models in which the scattering of two kinks produces a bound state [19], i.e., $|K_{ac}(\theta_1)K_{cb}(\theta_2)\rangle \to |K_{ab}(0)\rangle$, for some resonant rapidity $\theta_1 - \theta_2 = i\gamma$, $\gamma \in (0, \pi)$; this process is illustrated in Fig. 3 (a). The resonant rapidity gives the contact angle of the drop of phase $c$ adsorbed on the $a-b$ interface. The point where the three phases meet is the analogous – in $d = 2$ – of the three-phase contact line in three dimensions (see e.g., [47]). Since the contact point has to be in mechanical equilibrium, it follows that the surface tensions $\gamma_{ij}$ and the (dihedral) contact angles must be related. The condition for mechanical equilibrium at the three-phase contact is encoded in the relationship $\gamma_{ab} + \gamma_{ac} \cos \alpha + \gamma_{bc} \cos \beta = 0$. Alternatively, the equilibrium can be rephrased in terms of the Neumann triangle [47], which is shown in Fig. 3 (b). Recalling that the surface tension $\gamma_{ab}$ is identified in field theory with the lightest mass of the kink connecting the ground states $a$ and $b$, i.e., $\gamma_{ab} = m_{ab}$, the cosine formula – or al-Kashi's theorem – applied to the Neumann triangle yields $m_{ab}^2 = m_{ac}^2 + m_{cb}^2 + 2m_{ac}m_{cb} \cos \gamma$. The latter is the relativistic kinematic relation expressing energy conservation at the three-particle vertex.

It is intrinsic in the formalism that an imaginary Lorentz boost in the rapidity of the form $\theta \mapsto \theta + i\alpha$ corresponds to a particle forming an angle $\alpha$ with the $y$-axis. By applying this boost to boundaries[4] it is possible to rotate them and eventually a double boost can be employed to construct a wedge of opening angle $\pi - 2\alpha$. The mapping between observables for wetting at planar walls and wedge corners, with opening angle $\pi - 2\alpha$, involving the simple boost $\theta \mapsto \theta + i\alpha$, is termed *wedge covariance*. Such an elusive symmetry has been studied extensively within effective interfacial models [48–50] and its fundamental origin has been found in relativistic invariance of matrix elements [20].

Turing to correlation functions, it has been recently shown within field theory that the probabilistic picture applies also to the pair correlation function [28]. In this case we have

$$\partial_{x_1}\partial_{x_2}\langle\sigma(x_1, y_1)\sigma(x_2, y_2)\rangle_{ab} = (\langle\sigma\rangle_a - \langle\sigma\rangle_b)^2 P_2(x_1, y_1; x_2, y_2), \tag{6}$$

for the two-point function of the order parameter, and

$$\langle\varepsilon(x_1, y_1)\varepsilon(x_2, y_2)\rangle_{ab}^c \propto P_2(x_1, y_1; x_2, y_2), \tag{7}$$

---

[4]Purely reflecting tilted boundaries can be regarded as worldlines of particles with infinite mass.

Table 1: The dictionary establishing the correspondence between concepts in interfacial phenomena and field theory.

| Interfacial Phenomena | | Field Theory |
|---|:---:|---|
| 2D Euclidean plane | $\leftrightarrow$ | (1+1) Minkowski: Wick rotation |
| $(x, y)$ | $\leftrightarrow$ | $(x, t = iy)$ |
| rotation with angle $\alpha$ | $\leftrightarrow$ | imaginary Lorentz boost $\theta \rightarrow \theta + i\alpha$ |
| density field | $\leftrightarrow$ | spin field: $\sigma(x, y)$ |
| coexisting phases | $\leftrightarrow$ | ground states |
| $a - b$ interface | $\leftrightarrow$ | trajectory of the kink $K_{ab}(\theta)$ |
| surface tension $\gamma_{ab}$ | $\leftrightarrow$ | $m_{ab}$: mass of the lightest kink |
| contact angle $u$ at a flat wall | $\leftrightarrow$ | virtual rapidity $\theta = iu$ of surface bound state |
| mechanical equilibrium (Young's equation, etc.) | $\leftrightarrow$ | energy conservation $(P^0)$ |
| spreading coefficient $S = \gamma_{sv} - \gamma_{sl} - \gamma_{lv}$ [1] | $\leftrightarrow$ | $m_{lv}(\cos u - 1)$ |
| interfacial bifurcation | $\leftrightarrow$ | three-kink vertex (Fig. 3 (a)) |
| angle $\gamma$ of adsorbed bubbles | $\leftrightarrow$ | resonant rapidity (Fig. 3 (a)) |
| Neumann's triangle (Fig. 3 (b)) | $\leftrightarrow$ | relativistic kinematics for three kinks |
| wedge with opening angle $\pi - 2\alpha$ | $\leftrightarrow$ | boost $\theta \mapsto \theta \pm i\alpha$ of vertical walls |

for the two-point function of the energy density field. These two relations are the analogous, for pair functions, to (2) and (3), respectively. Here, $P_2(x_1, y_1; x_2, y_2)$ is the joint probability density of a Brownian excursion. The reconstruction of the profile at leading order in the large $R$ expansion occurs by inversion of the above relation, which yields a straightforward extension of (4). Analogously to one-point function, also for two-point functions field theory yields a systematic treatment of the finite-size corrections at order $R^{-1/2}$. These interface structure effects are $\propto \mathscr{A}$, where $\mathscr{A}$ is given by (A.16). We refer to [28] for a detailed account on this matter.

## 3  Models, geometry, and observables

Before presenting the results of our study, we provide a concise motivation about the choice of theoretical models. We consider the two-dimensional Ising model on the half-plane $x > 0$ with boundary conditions enforcing a droplet shape, as schematically illustrated in Fig. 1 (a). This model, which has been thoroughly studied both numerically and analytically, is the simplest lattice model able to reproduce the key phenomenological features of wetting and entropic repulsion [5]. Although the Ising universality class provides the minimal model for the study of interfacial phenomena and their interplay with bulk criticality, more complex models yielding richer wetting phenomena have been proposed [5]. The conceptual simplicity of the Ising model is due to the fact that below the critical temperature the system possesses only two degenerate ground states. This features suffices for the description of phase equilibrium in a

single component system, for instance, a liquid in coexistence with its vapour phase. Although the theoretical framework that we present in this paper applies to any 2D model exhibiting a second order phase transition and phase coexistence of at least two phases, the theory is tested against MC simulations for the Ising model.

Let $s_{i,j}$ be the Ising spin at site $(i,j)$. The simplest observable we will be dealing with is the order parameter profile, $\langle s_{i,j} \rangle_{-+}$, where the notation $\langle \dots \rangle_{-+}$ denotes expectation value with boundary the conditions inducing the droplet of Fig. 1. Since we will work within the framework of field theory, the order parameter profile will be denoted

$$\langle \sigma(x,y) \rangle_{-+}, \tag{8}$$

where $\sigma(x,y)$ is the spin field at the point with coordinates $(x,y) = (i,j)$. The measurement at a single point expressed by one-point functions typically suffices for the understanding of the phase separation phenomenon. For instance, the phenomenology of interfacial wetting leading to the splitting of a single interface to a double one is fully captured by the order parameter profile [32]. In this paper we will also investigate correlation functions between two spins, focusing in particular to

$$\langle \sigma(x,y)\sigma(x,-y) \rangle_{-+}. \tag{9}$$

The scope of our analysis is to clarify in a quantitative manner how these correlations depend on the separation between spins and on the distance from the wall when the latter is entropically repelling the interface or when it is undergoing wetting. We present the comparison between theory and simulations for entropic repulsion in Sec. 4 and then we will consider the wetting scenario in Sec. 5.

## 4 Entropic repulsion

### 4.1 Order parameter profile

Phase separation in Ising systems is studied by imposing a wall which plays the role of a substrate. The wall with patches of boundary conditions favoring different phases implements the formation of a droplet separating phases $a$ and $b$. For the Ising model with $a = -1$ and $b = +1$ this protocol introduces a droplet of negative magnetization enclosed in the Peierls contour indicated with solid green lines in Fig. 4 and whose definition is given on due course. When referring to the Ising model, we can alternatively rephrase results within the fluid-magnetic analogy. The mapping between variables in the lattice gas and Ising model is $n_{i,j} = (1+s_{i,j})/2$, where $n_{i,j} \in 0,1$ stands for an empty/filled site located at coordinates $(i,j)$ [6,24,51]. Without loss of generality, we can stipulate that the negative magnetization is the liquid phase and the positive magnetization is the vapour phase. When referring to the $a$-rich phase, we will simply term it as the liquid phase, while we will identify the vapour phase as the $b$-rich phase. In a real system this situation corresponds to a substrate with patch-wise inhomogeneous chemical composition, hydrophobic and hydrophilic.

In order to fix the notation we write the Hamiltonian of the Ising model with the boundary conditions described above,

$$\mathcal{H} = -J\sum_{i=1}^{l_x-1}\sum_{j=-l_y}^{l_y} s_{i,j}s_{i+1,j} - J\sum_{i=1}^{l_x}\sum_{j=-l_y}^{l_y-1} s_{i,j}s_{i,j+1} - J_b\sum_{j=-l_y}^{l_y} s_{0,j}s_{1,j}, \tag{10}$$

with spin variables $s_{i,j} \in \{-1, +1\}$ defined on a square lattice and the sum is restricted to nearest neighboring sites. In simulations, the lattice boundary is a rectangle[5] which comprises the sites $(i, j) \in [0, l_x - 1] \times [-l_y, l_y] \subset \mathbb{Z}^2$. The ferromagnetic interaction $(J > 0)$ between neighboring spins along the horizontal and vertical directions is described by the first and second terms respectively. The third term takes into account the (short-range) interaction between the wall $(i = 0)$ and the first column of spins $(i = 1)$. The strength of such an interaction is $J_b$. De facto, this wall-fluid interaction term is often written within the magnetic-fluid analogy in the form $-h_1(y)s_{1,j}$, so that $h_1(y) = J_b s_{0,y}$ is identified as the surface magnetic field [14]. For the study of the droplet shape we consider the patchwise boundary condition $s_{0,j} = \text{sign}(|j| - R/2)$, as sketched in Fig.4.

The system is studied for temperatures $T$ such that the bulk correlation length $\xi_b$ is much larger than the microscopic scale $a_0$ (lattice spacing in simulations) and $\xi_b \sim (T_c - T)^{-\nu}$ is much smaller than the separation $R$ between interface endpoints, i.e.,

$$a_0 \ll \xi_b \ll R, \tag{11}$$

the inequality $a_0 \ll \xi_b$ is needed for the continuum formalism to apply, while $\xi_b \ll R$ is needed for phase separation to emerge.

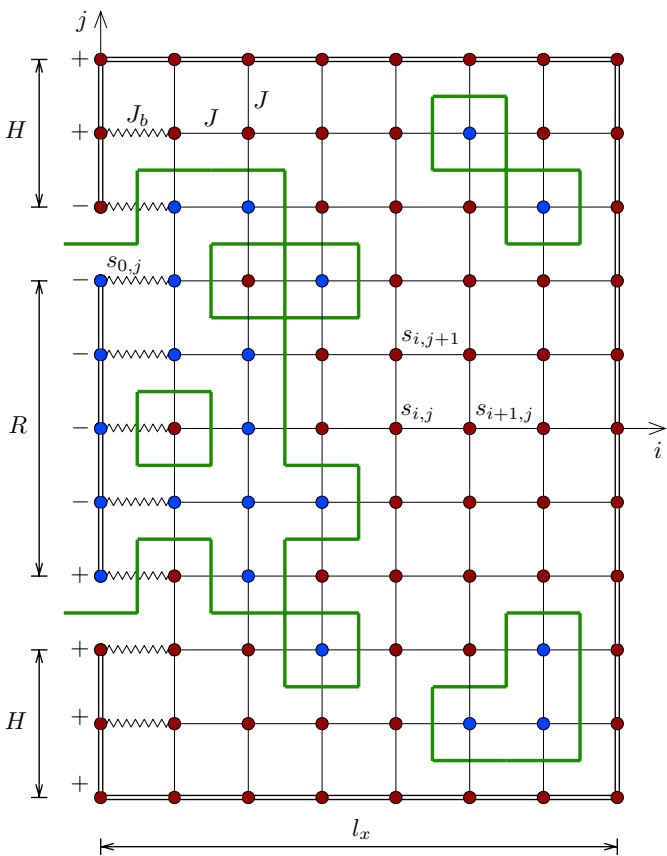

Figure 4: The setup employed in Monte Carlo simulations. Solid bonds have strength $J > 0$, the zigzags are weakened bonds of strength $J_b = \alpha J$. The double bonds along edges have infinite strength; hence, they connect spins with the same sign. Peierls contours are indicated with solid green lines on the dual lattice.

---

[5]The choice of the sizes $l_x$, $l_y$ is explained in the discussion about the Monte Carlo simulations at the end of this section.

In general, the coupling between (non-fluctuating) boundary spins and those in the adjacent layer is taken to be of the form $J_b = \alpha J$, where $0 \leqslant \alpha \leqslant 1$, meaning that $J_b$ is a weakened bond (these are represented with zigzags in Fig. 4). It is known that for $\alpha \in (0,1)$ there exist a wetting temperature $T_w(\alpha) < T_c$ such that $T_w(\alpha \to 0^+) = T_c^-$ and $T_w(\alpha \to 1^-) = 0^+$; we refer to [14] for the exact derivation of the wetting phase diagram for the square lattice Ising model. The scenario of entropic repulsion is expected for temperatures larger than the wetting one. To be definite, in our simulations for the case of entropic repulsion we simply take $J_b = J$, corresponding to $\alpha = 1$, meaning that for all subcritical temperatures the system is non-wet.

Let us consider a certain spin arrangement and draw a line (on the dual lattice) between each pair of opposite spins. This operation establishes a mapping from a spin configuration to a set of polygonal contours. These contours, hereinafter *Peierls contours*,[6] appeared in Peierl's proof of the spontaneous magnetization of the two-dimensional Ising model. For boundaries with all spins pointing in the same direction (all $+1$ or all $-1$) no contour can touch the boundary. For the case at hand, however, on top of closed polygonal loops there is an additional Peierls contour whose extremities are anchored at the two pinning points on the vertical wall, as depicted in Fig. 4 [53]. Beside isolated loops corresponding to bubble-like configurations, the remaining contours contribute to the droplet. In Sec. 4.2 we illustrate how to disentangle the contours branching off the droplet defining thus a non self-intersecting path connecting the pinning points. When bulk and boundary bonds in the Ising Hamiltonian have the same strength $J > 0$ the energy minimum for the Peierls contour at temperature $T = 0$ is attained by minimizing its length. However, for $T > 0$ the contour gains entropy by wandering in the semi-space $x > 0$, hence the entropic repulsion occurs [54]. The interface is pushed away from the wall but it cannot be expelled indefinitely far away from it because the contour is pinned to the boundary. As a result, the interface fluctuates as an elastic string with fixed endpoints. It is obvious that the volume enclosed by the droplet is not conserved. Parenthetically, we remark that is possible to investigate strongly fluctuating systems within the canonical ensemble corresponding to those instantaneous configurations with fixed total mass of the droplet. Interesting effects emerge both at the level of free energies [55] and correlations in interfacial phenomena [56].

Coming back to the droplet of Fig. 1 (a) and, as anticipated in Sec. 3, we pass to a continuum description in terms of fields. The average value of the spin $s_{i,j}$ with the boundary conditions enforcing the droplet is denoted as $\langle\sigma(x,y)\rangle_{-+}$, where $\sigma(x,y)$ is the spin field in the point $(x,y) = (i,j)$. Bulk and interfacial fluctuations build up a smooth density profile interpolating between the spontaneous bulk density $\langle\sigma\rangle_a$ inside the droplet and $\langle\sigma\rangle_b$ outside it. Expectation values in pure phases like $\langle\sigma\rangle_a$ are selected by taking spins fixed in state $a$ in a finite domain and then by removing the boundaries to infinity. The exact expression of the density profile for $x \gg \xi_b$ is [28]:

$$\langle\sigma(x,y)\rangle_{ab} = \frac{\langle\sigma\rangle_a + \langle\sigma\rangle_b}{2} - \frac{\langle\sigma\rangle_a - \langle\sigma\rangle_b}{2}\Upsilon(\chi) + \mathscr{A}P(x,y) + \mathcal{O}(R^{-1}). \qquad (12)$$

Some comments are in order. The quantities $\langle\sigma\rangle_a$ and $\langle\sigma\rangle_b$ depend on the system. In (12), $\Upsilon(\chi)$ is the universal scaling function

$$\Upsilon(\chi) = -1 - \frac{4}{\sqrt{\pi}}\chi e^{-\chi^2} + 2\,\mathrm{erf}(\chi), \qquad (13)$$

the latter does not depend on the system in question provided the phases $a$ and $b$ can be separated through a single interface.[7] The coordinates $x, y$ enter via $\chi = x/(\kappa\lambda)$ where

---

[6]Originally termed boundary lines in [52].

[7]Phase separation via double interfaces has been studied in [19] and the necessary conditions for intermediate phases have been classified in [90].

$\lambda = \sqrt{R\xi_b}$, and $\kappa = \sqrt{1-(2y/R)^2}$ is a $y$-dependent rescaling factor. Leaving aside the term $\mathscr{A}P(x,y)$, it is simple to show how the density profile reaches the bulk value $\langle\sigma\rangle_b$ for $x \gg \lambda$; the parameter $\lambda \sim R^{1/2}$ sets the length scale associated to interfacial fluctuations in the $x$ direction. Although the function $\Upsilon(\chi)$ appeared in the literature of Ising interfaces [14, 18, 57–59], the term $\mathscr{A}P(x,y)$, which has not been found in previous analyses, is a *novel* result.

Let us discuss the subsequent term appearing in the density profile. The quantity $P(x,y)$ is the probability density to find the interface in the point $(x,y)$ while $\mathscr{A}$ is a factor that depends on the specific universality class. The probability for the interface to cross the horizontal axis ($y=0$) in the interval $(x, x+dx)$ is $P(x,0)dx$, where

$$P(x,0) = \frac{4x^2}{\sqrt{\pi}\lambda^3}e^{-x^2/\lambda^2}, \tag{14}$$

while for $|y| < R/2$ the corresponding result is $P(x,y) = \kappa^{-1}P(x/\kappa, 0)$. It has to be noticed that $P(x,y)$ vanishes along the wall. This is a signature of the entropic repulsion experienced by the interface which is encoded in the factor $x^2$ in (14). There is another remark to point out: while $\Upsilon(\chi)$ and $P(x,y)$ are *super-universal*, i.e., they are shared by several universality classes (e.g., Ising, Potts, etc [28]), the amplitudes $\langle\sigma\rangle_a$, $\langle\sigma\rangle_b$ and $\mathscr{A}$ depend on the universality class of the model.[8] For the Ising model, which is the system we examined in simulations, $\mathscr{A} = M/\gamma_{lv}$ and $\gamma_{lv}$ is the surface tension (in $k_B T$ units) of the $a-b$ interface, i.e., the analogous of te liquid-vapour interface. The surface tension is related to the subcritical correlation length via

$$\gamma_{lv}\xi_b = \frac{1}{2}. \tag{15}$$

Although this relation is derived within the field-theoretical framework that applies in the scaling region near criticality, for the Ising model the identity (15) is an exact form of Widom's relation valid for all subcritical temperatures, a result that follows from duality arguments [13, 60, 61]; see also [62, 63]. It follows from (14) that $\mathscr{A}P(x,y)$ yields a correction proportional to $R^{-1/2}$ times some function of $\chi$, therefore it provides a finite-size correction to the density profile. In general, these corrections can be organized in the form of a series expansion in the small parameter $\sqrt{\xi_b/R}$ and the term $\propto \mathscr{A}$ is the first one of such an expansion.

It follows from (12) that the leading form of droplet shapes – i.e., when $R$ is large enough to neglect the interface structure effect – is given by the isolines of the density profile $\Upsilon(\chi)$. Since the latter depends on $x$ and $y$ via $\chi$, the isolines are arcs of ellipses, as illustrated with the solid white lines in Fig. 1 ($b$), with the solid black line as the isoline at zero value. This theoretical prediction is confirmed by MC simulations carried out for both Ising and Potts models [64]. Upon including the interface structure correction the isoline shift, this is shown with the dashed black line in Fig. 1 ($b$) corresponding to the shift of the solid black line. Alternatively, the droplet shape can be also characterized as the locus in the $(x,y)$ plane in which the passage probability $P(x,y)$ is maximum or, similarly, as the average height with respect to the wall. The average height, which is given by (28), is indicated in Fig. 1 ($b$). It is obvious that all these definitions are equivalent up to an overall numerical coefficient which is of order $\mathcal{O}(1)$.

We are now in the position to apply the general result for the density profile [Eq. (12)] to the specific case of the Ising model and confront the corresponding outcome against Monte Carlo simulations. For the Ising model the global $\mathbb{Z}_2$ spin reversal symmetry implies that $\langle\sigma\rangle_\pm = \pm M$ where $M$ is the spontaneous magnetization. This symmetry is spontaneously broken for temperatures below the critical one given by $k_B T_c/J = 2/\ln(1+\sqrt{2}) \approx 2.269$ [23, 65]. The exact expression for the spontaneous magnetization has been calculated by C. N. Yang in

---

[8]The factor $\mathscr{A}$ – as well as the scaling function $\Upsilon$ – depends also on the geometry of the system, e.g., it takes different values for strip and the half-plane [28]. See also Sec. 7 for additional remarks on this point.

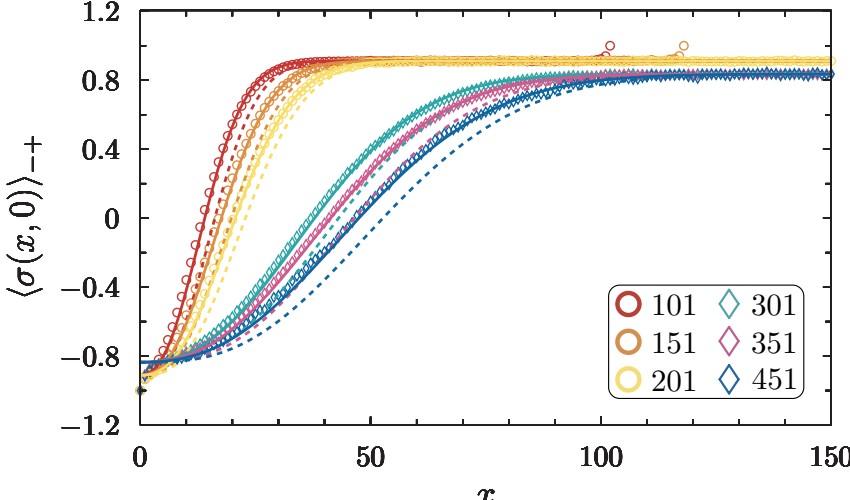

Figure 5: Density profiles for $T = 2$ (circles) and $T = 2.15$ (diamonds) and $R$ shown in the inset. The exact prediction including the subleading correction proportional to $R^{-1/2}$ [Eq. (12)] is indicated with solid lines. Dashed lines indicate the magnetization profile without the subleading correction. Error bars are not indicated but typically their extent does not exceed that of the symbols in this figure and subsequent ones.

1952 [66] and reads $M = \left(1 - (\sinh(2K))^{-4}\right)^{1/8}$, where $K \equiv J/(k_B T)$ is the reduced coupling and $k_B$ is Boltzmann's constant. The bulk correlation length is given by $\xi_b = (4K - 4K^\star)^{-1}$ [13], where $K^\star$ is the dual coupling defined by $\exp(-2K^\star) = \tanh K$. For completeness we provide the corresponding expressions in the scaling limit: $M \sim m_0 (T_c - T)^\beta$, $\beta = 1/8$ and $\xi_b \sim \xi_0^- (T_c - T)^{-\nu}$, $\nu = 1$, with critical amplitudes $\xi_0^- = T_c^2/8$ and $m_0 = 2^{7/16} T_c^{-1/4}$ [67,68]. However, since both $M$ and $\xi_b$ are known for any temperature, we used their exact analytic expressions. Without loss of generality, from now on we will set $k_B = 1$ and $J = 1$ and all lengths will be measured in units of the lattice spacing.

Early numerical simulations[9] for an Ising droplet in 2D exhibited discrepancies when compared against the leading-order profile [42] $\Upsilon(\chi)$. The correction term $\mathscr{A}P(x, y) \propto R^{-1/2}$, which we have identified, turns out to resolve the aforementioned mismatch and eventually yields – if included in the profile – an accurate comparison between theory and numerics, as shown in Fig. 5. The agreement is perfect due to such a term and without taking it into account the MC data fall systematically away from the analytic prediction. The small deviations for $x \to 0$ are due to the fact that the exact result does not take into account wall effects which fade away only for $x \gg \xi_b$; see [70] about exact calculations for the square lattice Ising model. Deviations for large $x$ in Fig. 5 are simply due to the finiteness of the simulation box. For the sake fo completeness we point out that MC simulations of Ref. [42] were performed for $T \approx 1.9$ and $R \approx 46$. In this work, we examined larger system sizes ranging from $R = 151$ to $R = 451$ and temperatures $T = 2$, $T = 2.1$, and $T = 2.15$, which are even closer to the critical point ($T_c = 2.26919...$). In the domain of applicability of the theoretical result, which is $x \gg \xi_b$, the analytic expression (12) and the data points from simulations are superimposed with good accuracy.

Before concluding this section we summarize some details about the Monte Carlo simulations. Quite similarly to our previous studies [30,31,71], we performed Monte Carlo calculations by using a hybrid scheme which combines the standard Metropolis algorithm (see, e.g. [72]) and the Wolff cluster algorithm [73]. Parallelization has been achieved by inde-

---

[9]See [69] for MC studies of interfaces in Ising films and [12] for a review on simulations.

pendently and simultaneously simulating up to 360 Ising lattices on a parallel computer. An appropriately seeded family of dedicated, very large period, Mersenne Twister random number generators [74] (in the MT2203 implementation of the Intel Math Kernel Library) was used in order to simultaneously generate independent sequences of random number to be used for the MC updates of the lattices. Each independent parallel simulation comprised a warm up phase in which $10^9$ numbers drawn from each random number stream in order to warm up the generators followed by a thermalization of the lattice of $3 \cdot 10^6 \times N$ Metropolis steps on random spins, with $N$ the total number of spins in the simulation box. Every production step consisted on $10^7$ hybrid updates consisting a Wolff cluster update at a random point in the lattice and $[N/2]$ Metropolis spin flips on random spins, $[\dots]$ stands for the integer part. The simulation box is a rectangle comprising $N = l_x(2l_y + 1)$ spins and the vertical size is $l_y = H + (R-1)/2$, with $R$ an odd integer. The distance $H$ is chosen such that $H/\lambda \gtrsim 4$ while $l_x/\lambda \gtrsim 7$. This design effectively reduces the influence of the boundaries and allow simulations in a finite box to be meaningfully compared against the field-theoretical result derived on the half-plane.

## 4.2 Passage probability

The comparison between theory and simulations provided in Fig. 5 corroborates the necessity of the interface structure correction $\propto P(x, y)$. However, it would be optimal to test the theoretical prediction (14) in a direct fashion even without detecting the order parameter profile. This is what will be discussed in this section. The idea is to extract $P(x, y)$ directly by collecting statistics of lattice-defined interfacial shapes.

The interface on the square lattice can be constructed in several ways. One way is to draw a segment on the dual lattice for bonds which connect opposite spins [53]. A schematic picture of the resulting interface is illustrated with the green curve (continuous and dashed) in Fig. 6. In general, it is possible to regard the interface as the result of an exploration process which starts from one pinning point and ends at the other; see points denoted "*in*" and "*out*" in Fig. 6. The process constructed on the dual lattice is however ambiguously defined in correspondence of those plaquettes in which the interface splits in three, as indicated with dashed lines in Fig. 6. The occurrence of plaquettes in which the exploration process is not uniquely defined causes the formation of loops and overhangs.

From the viewpoint of simulation studies it is crucial to employ a definition of interface that allows for tracking without ambiguities induced by the formation of branching and loops. In order to remove such an ambiguity it is possible to employ the definition of interface on the *medial lattice* [75]. Interfaces on the medial lattice are always well defined and do not self-intersect; see blue and red curvy lines in Fig. 6. Alternatively, the loops represented by dashed green bonds can be removed by defining the interface as a loop erased random walk (LERW) [75]. The result of this procedure is the solid line in Fig. 6 which connects the two pinning points. As can be seen in Fig. 6, the pinning points emanate two interfaces on the medial lattice with the LERW-defined interface enclosed in between.

The passage probability $P(x, 0)$ is obtained from MC simulations by sampling the number of crossings at different intervals on the lattice. The number of crossings in each interval is used in order to construct a histogram, which thus represent a discretized version of $P(x, 0)$. The result of this analysis is summarized in Fig. 7. The theoretical result is found to be in good agreement with the MC simulations without fitting parameters. As emphasized, overhangs are inevitably observed in MC simulations and, as a consequence, multiple crossings of the horizontal axis are generated. To this end it is necessary to give a certain prescription which deals with multiple crossings, or to consider the sampling over the ensemble of single crossings. In our simulations, we have first traced the interface on the lattice by restricting the statistics to single crossings. We have observed that defining the interface with the LERW or the medial lattice construction does not provide appreciable deviations. The remarkable agreement between theory and MC

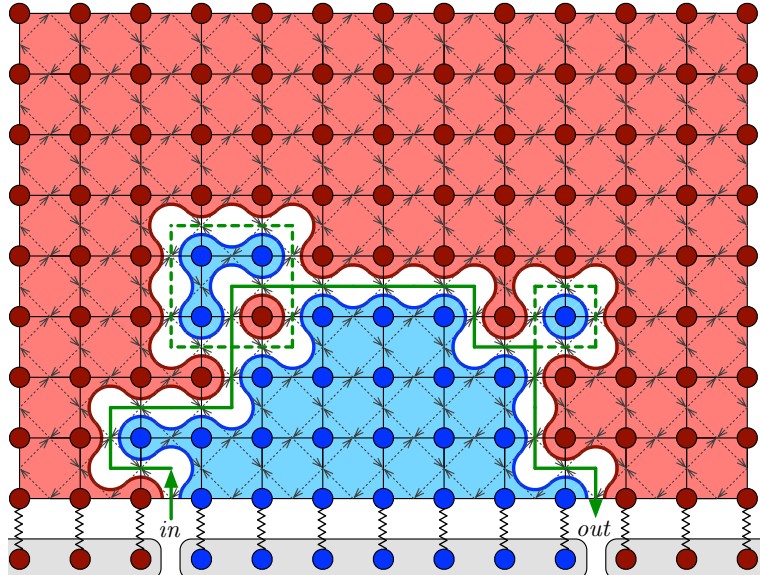

Figure 6: The interface for the square lattice Ising model seen as an exploration process starting in the point denoted in and terminating in the point denoted out. Green bonds denote the interface on the dual lattice. Curvy lines are the cluster boundaries on the medial lattice. Bulk spins fluctuate while boundary spins (in gray blocks) are fixed. Bulk bonds and zigzag ones have strength $J > 0$ in our simulations for entropic repulsion effects.

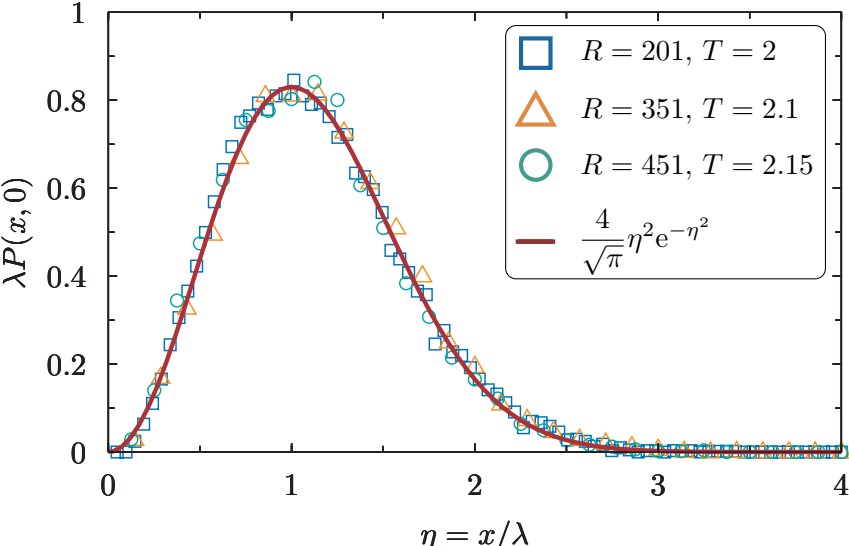

Figure 7: Rescaled passage probability density $\lambda P(x, 0)$ as a function of the rescaled horizontal coordinate $\eta = x/\lambda$. Symbols are obtained from MC simulations, the theoretical result [Eq. (14)] is indicated with the solid red line.

data at different $T$ and $R$ corroborates the validity of the field theory for off-critical interfaces presented in this paper.

Parenthetically, we stress that exact results for strongly fluctuating off-critical interfaces in the 2D Ising model (see D. B. Abraham's review [13]) were used to test the reliability of heuristic description in terms of random walks (see M. E. Fisher's review [54]). In particular, the fact that midpoint fluctuations of the interface grow as the square root of the separation between pinning points has been rigorously proved for low temperatures by G. Gallavotti long time ago [76]. Then, the convergence of interface fluctuations towards the Brownian bridge has been rigorously proved in more recent times for the Ising model [77] and for the $q$-state Potts model [78].

The occurrence of Brownian bridges in the description interfacial phenomena is at the core of several studies. We mention a few of them beyond the field-theoretical results of [18,19,28]. Scaled Brownian bridges appear in the rigorous formulation [79] of fluctuations for an inclined interface in the 2D Ising model that have been examined within exact transfer matrix methods in [80]. Brownian bridges also appear in studies of prewetting (see, e.g., [81]) in the 2D Ising model [82], rigorous studies of dynamics in 2D Ising model [83], and recent investigations on the Cahn-Hilliard free energy derived within the Bethe-Guggenheim approximation [84].

The technique of interface tracking we illustrated in this paper has been predominantly[10] applied in the context of *critical interfaces*, i.e., those random curves describing fluctuating cluster boundaries emerging in lattice models *at* the critical point. Contrary to the off-critical interfaces examined in this paper, due to the infinite correlation length critical interfaces exhibit the higher symmetry of conformal invariance and admit a mathematical treatment in terms of Schramm Loewner Evolution. The reader interested on interfaces at the critical point can find pedagogical review material in [75, 86].

## 4.3 Correlations

The density-density correlation function in direction parallel to the wall reads [28]

$$\langle \sigma(x,y)\sigma(x,-y)\rangle_{ab} = \frac{\langle\sigma\rangle_a^2 + \langle\sigma\rangle_b^2}{2} + \frac{\langle\sigma\rangle_a^2 - \langle\sigma\rangle_b^2}{2}\Upsilon(\eta) - \frac{4}{\pi}(\langle\sigma\rangle_a - \langle\sigma\rangle_b)^2\sqrt{\frac{2y}{R}}\,\eta^2 e^{-\eta^2}$$
$$+ \mathcal{O}((y/R)^{3/2}), \tag{16}$$

where $\eta = x/\lambda$ is the rescaled distance from the wall. For the sake of simplicity of exposition we provide a result that applies to small vertical separations such that $\xi_b \ll y \ll R$. An expression valid for arbitrary positions of the two fields has been elaborated in [28], including corrections proportional to $1/\sqrt{R}$. Recalling that interfacial fluctuations in the presence of a wall exerting entropic repulsion are described in terms of Brownian excursions, the result (16) is nothing but a joint cumulative distribution function of a Brownian excursion.

The comparison against MC simulations is shown in Fig. 8 for the connected correlation function, $G_{\parallel}^c(x,y) = \langle\sigma(x,y)\sigma(x,-y)\rangle_{ab} - \langle\sigma(x,y)\rangle_{ab}\langle\sigma(x,-y)\rangle_{ab}$. The term proportional to $\sqrt{y}$ in Eq. (16) is the signature of long-range correlations. The factor $\eta^2$ – responsible for entropic repulsion – penalizes correlations in proximity of the wall, while the Gaussian term $\exp(-x^2/R\xi_b)$ suppresses correlations far from the wall over the length $\sqrt{R\xi_b}$. The combined effect along parallel and perpendicular directions is the *confinement* of long-range correlations in the interfacial region, a feature pointed out long time ago [15] but the analytic form of these correlations has never been derived before.

---

[10]See [85] for a simulation study at and away from bulk criticality.

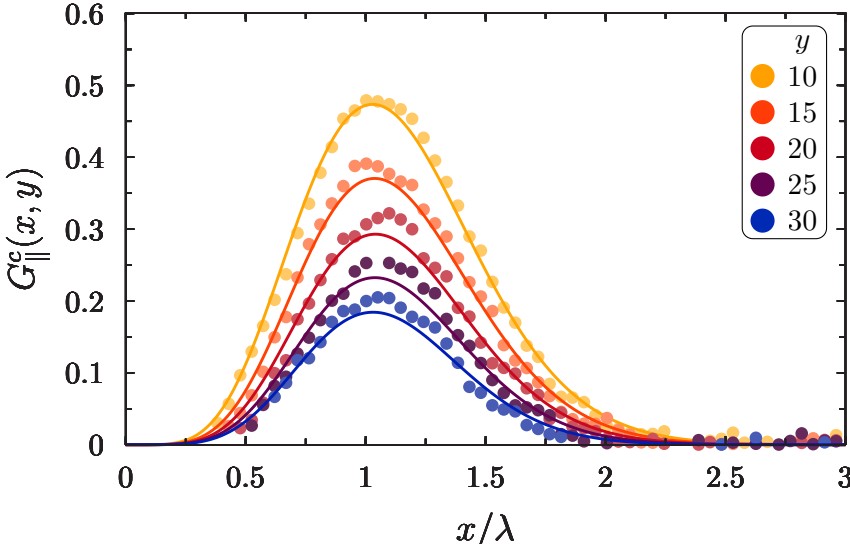

Figure 8: The connected parallel correlation function in units of the squared bulk magnetization $M^2$. Data points are obtained from MC simulations of the Ising model at $T = 2.0$ and $R = 201$ with $y$ reported in the inset.

## 5 Wetting transition

### 5.1 Phenomenology and results from field theory

In the partially wet state a sessile droplet on a flat substrate is characterized by a contact angle $\Theta$ satisfying Young-Laplace equation [1, 47]

$$\gamma_{sv} = \gamma_{sl} + \gamma_{lv} \cos\Theta, \tag{17}$$

where $\gamma_{sv}, \gamma_{sl}$ are the surface tensions of the *solid-vapour* and *solid-liquid* interface, respectively, and $\gamma_{lv}$ is the surface tension of the *liquid-vapour* interface; see Fig. 9 (a). Beside the picture of surface unbinding mentioned in the introduction, wetting transitions can be formulated as

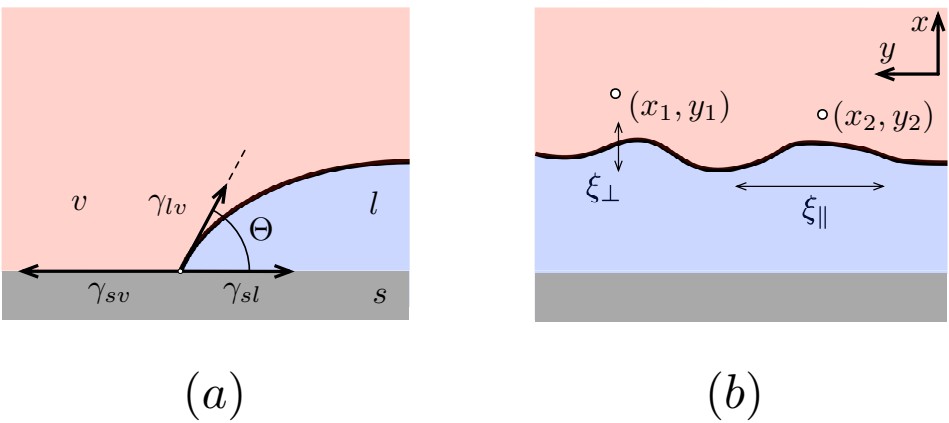

Figure 9: Panel (a): a sessile droplet separates a liquid ($l$) in equilibrium with its vapour ($v$) phase forming a contact angle $\Theta$ on a solid wall ($s$). Panel (b): parallel and perpendicular correlation lengths for an infinitely long interface.

the process in which the contact angle $\Theta$ vanishes as $T$ approaches $T_w^-$. When $\Theta = 0$ the surface tensions satisfy Antonov's rule $\gamma_{sv} = \gamma_{sl} + \gamma_{lv}$ [47, 104]; we refer to Appendix A for a discussion about this point in connection with entropic repulsion. The mechanism of interface binding manifests in the field-theoretic formalism as a boundary bound state between the wall and the particle excitation associated to the interface [87]; see Appendix A. The interaction between the wall and interfaces is accounted for by a matrix element that exhibits a pole-type singularity of the form $1/(\theta - iu)$ for $\theta \to iu$, where $\theta$ is the rapidity of relativistic particles associated to fluctuating interfaces and, as in Sec. 4.1, $m = \gamma_{ab}$ is the surface tension of the $a-b$ interface;[11] i.e., $p = m \sinh\theta$ is the momentum, $m \cosh\theta$ is the energy. A simple energy-balance argument implies that $e_B' = e_B + m \cosh\theta_0$ is the bound state energy, with $e_B$ the energy of the system when the boundary is uniform. The virtual rapidity $\theta_0 = iu$ characterizing the bound state leads to the Young-Laplace equation (17) with the identification $e_B' = \gamma_{sv}$, $e_B = \gamma_{sl}$ and the contact angle in the macroscopic model is given by the location of the wetting pole; hence, $\Theta = u$.

One- and two-point correlation functions for temperatures close to the wetting transition can be calculated within the framework of the exact field theory of phase separation [18]. One-point functions of both the spin field and energy density are computed in Appendix A and the results are summarized in Sec. 5.2. The calculation of two-point correlation functions is technically more involved but nonetheless the mathematical determination of these quantities follows the same strategy adopted for one-point functions. The result for the density-density correlation is given in Sec. 5.3 and its calculation is sketched in Appendix B.

## 5.2 Order parameter profile

The density profile is given by

$$\langle\sigma(x)\rangle_{ab} \approx \langle\sigma\rangle_b + (\langle\sigma\rangle_a - \langle\sigma\rangle_b)e^{-2x/\xi_\perp}, \tag{18}$$

where $\xi_\perp^{-1} = mu$ is the perpendicular correlation length and for the Ising model $u = \Theta \sim T_w - T$ [20] measures the departure from the wetting temperature. As expected, there is no $y$-dependence in the profile (18). The translational invariance along the direction parallel to the wall follows since we are dealing with an infinitely long interface. From (18) it follows that the passage probability for the wetting regime is $P^{(w)}(x) = (2/\xi_\perp)e^{-2x/\xi_\perp}$; we refer to Sec. 2 and in particular to (4) for details about the probabilistic interpretation. The thickness $\ell$ of the wetting layer is defined as the average distance of the interface from the wall, hence it is given by the first moment of $P^{(w)}(x)$, i.e $\ell = \int_0^\infty dx\, x P^{(w)}(x)$. A simple calculation gives $\ell = (\xi_\perp/2) \sim (T_w - T)^{-\beta_s}$. For the Ising model the scaling law $\xi_\perp \sim (T_w - T)^{-\nu_\perp}$ is satisfied with critical exponent $\nu_\perp = 1$. The latter gives for the thickness of the wetting layer $\ell \sim (T_w - T)^{-\beta_s}$, with surface critical exponent $\beta_s = 1$, in agreement with path-integral formulations [58].

## 5.3 Correlations

Let us consider the pair function of the energy density field $\varepsilon(x, y)$. It can be shown that

$$\langle\varepsilon(x_1, y_1)\varepsilon(x_2, y_2)\rangle_{-+}^c \propto P_2^{(w)}(x_1, x_2; y_1 - y_2), \qquad y_1 > y_2, \tag{19}$$

in the above, $P_2^{(w)}(x_1, x_2; y_1 - y_2)dx_1 dx_2$ is the net probability for the interface to cross the interval $(x_1, x_1 + dx_1)$ *and* $(x_2, x_2 + dx_2)$ separated by the distance $y_1 - y_2$, as shown in Fig. 9 (b).

---

[11] See Sec. 2 for an overview on the theory.

The analytic expression of $P_2^{(w)}$ is provided in Appendix B. As a matter of fact, $P_2^{(w)}$ depends only on the rescaled coordinates

$$X_j = x_j/\xi_\perp, \qquad Y = (y_1 - y_2)/\xi_\parallel, \tag{20}$$

where $\xi_\perp = 1/(mu)$ and $\xi_\parallel = 2/(mu^2)$, where $m$ is the surface tension of the interface (in $k_B T$ units). Calculation entails

$$P_2^{(w)}(x_1, x_2; y_1 - y_2) = \xi_\perp^{-2} \Pi_2(X_1 - X_2, X_1 + X_2; Y), \tag{21}$$

with $\Pi_2(X, \bar{X}; Y)$ given by (B.3). The connection between energy density correlations and passage probabilities at wetting has been established long time ago for the Ising model by means of exact calculations on the square lattice [88] and in [58] for Solid-On-Solid models. The result we derived applies, more generally, to those models which exhibit the wetting phenomenon – i.e., a bound state pole appearing in the boundary $S$-matrix –, with the Ising model as the simplest representative [87].

Focusing on the Ising model below wetting temperature, the parallel correlation function in direction parallel to the wall is

$$\langle \sigma(x,y)\sigma(x,-y)\rangle_{-+} = M^2 \left[ -1 - 4e^{-2X}\,\text{erf}\left(\sqrt{Y}\right) + 2\,\text{erf}\left(\frac{Y+X}{\sqrt{Y}}\right) + 2e^{-4X}\,\text{erfc}\left(\frac{X-Y}{\sqrt{Y}}\right) \right], \tag{22}$$

where $X = x/\xi_\perp$ and $Y = y/\xi_\parallel$. The plot of the density-density correlation function for arbitrary $X_1$ and $X_2$ is shown in Fig.10. The result for $X_1 = X_2$, corresponding to the parallel correlation function, is indicated with the solid black curve in Fig.10. For small separations in direction parallel to the wall such that $y \ll \xi_\parallel$ the result (22) can be expanded as follows

$$\langle \sigma(x,y)\sigma(x,-y)\rangle_{-+} = M^2 - 8M^2\sqrt{\frac{2y}{\pi\xi_\parallel}}\,e^{-\frac{2x}{\xi_\perp}} + \mathcal{O}(y^{3/2}), \tag{23}$$

up to terms of order $\mathcal{O}((y/\xi_\parallel)^{3/2})$. This expansion shows how the points $(x, y)$ and $(x, -y)$ are indeed long-range correlated because the correlation approaches the bulk value $M^2$ with a power-law term proportional to $\sqrt{y/\xi_\parallel}$. In analogy with entropically repelled interfaces (see Eq. (16)), $\sqrt{y}$ is the signature of long-ranged correlations. From (22) we can directly check the clustering property of correlation functions, e.g.,

$$\lim_{x_2 \to +\infty} \langle \sigma(x_1, y_1)\sigma(x_2, y_2)\rangle_{-+} = \langle \sigma(x_1, y_1)\rangle_{-+}, \tag{24}$$

the latter is visualized with dashed green lines in the plot of Fig. 10.

The structure of (16) and (23) are strikingly similar to each other, apart from two differences that is worth highlighting. Firstly, the entropic repulsion term $\propto x^2$ – which multiplies the exponential envelope in (16) – does not occur in (23). Secondly, the result obtained for entropic repulsion can be brought in the form (23) provided one identifies the parallel and perpendicular correlation lengths with $\xi_\parallel^{(er)} = R$ and $\xi_\perp^{(er)} = \lambda = \sqrt{R\xi_b}$. This identification allows us to write, for $R \to \infty$, $\xi_\perp^{(er)} \sim (\xi_\parallel^{(er)})^\zeta$ with the wandering exponent $\zeta = 1/2$, which is the well-known exponent for interfaces driven by thermal disorder in two dimensions.

## 6 Interface structure factor

The starting point towards a definition of ISF for the droplet commences with the introduction of a connected pair correlation function $G_\parallel^{\text{conn.}}$ which carries *only* interfacial degrees of freedom

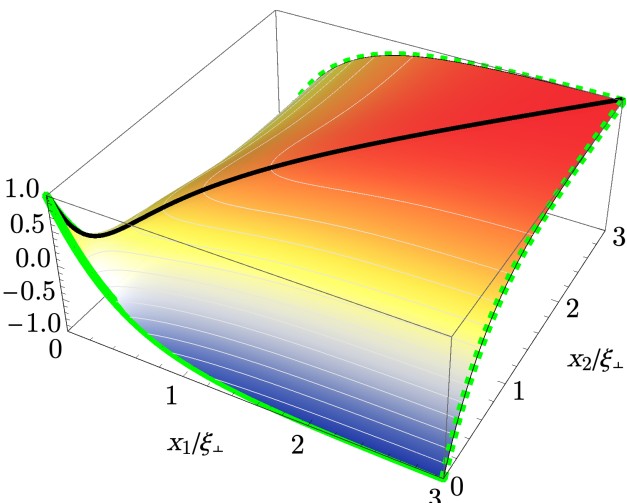

Figure 10: The density-density correlation function $\langle \sigma(x_1, y_1)\sigma(x_2, y_2)\rangle_{-+}/M^2$ below the wetting temperature as a function of the rescaled distances from the wall $x_j/\xi_\perp$. In this figure $(y_1 - y_2)/\xi_\parallel = 0.1$ and the analytic result is given by (B.8). The solid black line obtained for $x_1 = x_2$ corresponds to the parallel correlation function [Eq. (22)]. The dashed green lines are the density profile $(1 - 2\exp(-2X_1))$ and $(1 - 2\exp(-2X_2))$ obtained either for large $x_2$, and $x_1$, respectively, these are obtained from the clustering property (24).

and not bulk ones. The natural way of proceeding is to define

$$G_\parallel^{\text{conn.}}(x_1, x_2; y) = \langle \sigma(x_1, y)\sigma(x_2, -y)\rangle_{ab} - \langle \sigma(x_1, y)\rangle_{ab}\langle \sigma(x_2, -y)\rangle_{ab} - G_{\text{bulk}},$$

where $G_{\text{bulk}} = \langle \sigma(x_1, y)\sigma(x_2, -y)\rangle_b - \langle \sigma(x_1, y)\rangle_b\langle \sigma(x_2, -y)\rangle_b$ is the two-point function in the phase outside the droplet, the $b$-phase. This prescription removes bulk correlations when spin fields are infinitely far from the interface but at finite separation between them [22].

Let us consider first the case of entropic repulsion from the hard wall. Since the interface extends from $y = -R/2$ to $y = R/2$, the ISF $S(q)$ is defined as the *finite* parallel Fourier transform of the integrated connected correlator, something that we write as follows

$$S(q) = \frac{1}{2}\int_{-R/2}^{R/2} dy\, e^{iqy} \iint_0^{+\infty} dx_1 dx_2\, G_\parallel^{\text{conn.}}(x_1, x_2; |y|). \tag{25}$$

It can be shown that integrating the connected density-density correlation function is equivalent, up to a proportionality factor, to the integral of the height-height correlation function $\mathscr{C}(|y|)$ given by

$$\mathscr{C}(y) = \overline{h(y)h(-y)} - \overline{h(y)}\,\overline{h(-y)}. \tag{26}$$

The proportionality factor is the squared jump of density across the interface, the latter reads $(\Delta\langle\sigma\rangle)^2 \equiv (\rho_l - \rho_v)^2$. Hence, the interface structure factor can be equivalently expressed as

$$S(q) = \frac{(\rho_l - \rho_v)^2}{2}\int_{-R/2}^{R/2} dy\, e^{iqy}\mathscr{C}(|y|). \tag{27}$$

For the regime of entropic repulsion the average distance of the interface from the wall reads

$$\overline{h(y)} = \frac{2}{\sqrt{\pi}}\sqrt{R\xi_b}\sqrt{1 - (2y/R)^2}, \tag{28}$$

a result that follows by computing the first moment of the Brownian excursion. As expected, $\overline{h(y)}$ is symmetric under reflections about the $x$-axis and it vanishes in the pinning points $y = \pm R/2$. The most probable interface location is thus proportional to $\overline{h(y)}$ and yields the solid black arc of ellipse shown in Fig. 1 ($b$). The height-height correlation function is the covariance of the Brownian excursion; hence,

$$\mathscr{C}^{(\text{er})}(y) = \frac{4}{\pi} R \xi_{\text{b}} (1 - \tau^2) \big[ V(\tau) - 1 \big], \tag{29}$$

the above is expressed in terms of Gauss' hypergeometric function

$$V(\tau) = {}_2F_1\left(-\tfrac{1}{2}, -\tfrac{1}{2}, \tfrac{3}{2}; (1-\tau)^2/(1+\tau)^2\right). \tag{30}$$

By plugging (29) into (27) we obtain the interface structure factor for the regime of entropic repulsion. Focusing on the physical aspects, we observe how the wave-vector $q$ has to be much larger than the lower momentum cutoff $\Lambda = 1/R$ imposed by the system size. Analogously, $q$ cannot exceed the upper momentum cutoff set by the inverse bulk correlation length, which plays the role of a microscopic scale. Consequently, the following result is obtained for $R^{-1} \ll q \ll \xi_{\text{b}}^{-1}$:

$$S(q) \simeq \frac{(\Delta \langle \sigma \rangle)^2}{\gamma_{lv} q^2} \left[ 1 - \frac{32}{\sqrt{\pi}} \frac{1}{(qR)^{3/2}} + \cdots \right], \tag{31}$$

up to corrections due to the interface structure $\propto \mathscr{A}$ which have been computed in [28]. Notably, the underlying entropic repulsion affects the ISF with the term enclosed in square brackets. This feature is absent in the ISF of an interface on the strip because entropic effects in such a case do not occur and the capillary-wave result, $S(q) \approx (\rho_l - \rho_v)^2/\gamma_{lv} q^2$, is obtained up to finite-size corrections of order $\mathcal{O}(1/R)$ [22]. We recall that the vicinity of bulk criticality $\gamma_{lv} \sim (T_c - T)^\mu$, therefore the overall factor $(\rho_l - \rho_v)^2/\gamma_{lv}$ scales as $M^2/m \sim (T_c - T)^{-\omega}$ with the critical exponent $\omega = \mu - 2\beta$. Equivalently, we can use Widom's hyperscaling relation $\mu = (d-1)\nu$ [10] to express, in $d = 2$, $\omega = \nu - 2\beta$. The signature of entropic-repulsion effects – which in real space are encoded in the term $y^{1/2} x^2 \exp(-x^2/R\xi_{\text{b}})$ in the pair correlation function [Eq. (16)] – appear in momentum space with the term enclosed in square brackets in (31).

The wetting scenario can be analyzed *mutatis mutandis*. For an infinitely long interface ($R \to \infty$) the integration over the parallel coordinate $y$ extends over the real line and the appropriate correlation function has to be used. As shown in Appendix C, the connected height-height correlation function at wetting

$$\mathscr{C}^{(\text{w})}(y_1 - y_2) = \overline{h(y_1)h(y_2)} - \overline{h(y_1)}\ \overline{h(y_2)}, \tag{32}$$

is translational invariant, i.e., it depends only on $|y_1 - y_2|$ and is given by

$$\mathscr{C}^{(\text{w})}(y_1 - y_2) = \frac{\xi_\perp^2}{4} \mathcal{I}((y_1 - y_2)/\xi_\parallel), \tag{33}$$

where $\mathcal{I}(Y)$ is the function

$$\mathcal{I}(Y) = \frac{2}{\sqrt{\pi}}(1 + 2Y)\sqrt{Y}\,\text{e}^{-Y} - (4Y^2 + 4Y - 1)\text{erfc}\left(\sqrt{Y}\right). \tag{34}$$

Deferring to Appendix C for the details of the calculation, the interface structure factor reads

$$S(q) = \frac{(\rho_l - \rho_v)^2}{\gamma_{lv} q^2} \left[ 1 - \sqrt{2}\left(1 + \sqrt{1 + q^2 \xi_\parallel^2}\right)^{-1/2} \right]. \tag{35}$$

Notably, as shown in Appendix C, the parallel and perpendicular correlation length appear in front of the $q^2$ term with the combination $\xi_{\parallel}/2\xi_{\perp}^2 = \gamma_{\mathrm{lv}}$, which coincides with the surface tension of the interface. As a result, the large-$q$ asymptotic behavior, which is captured by the term in front of the square bracket in (35), is the expected form of $S(q)$ for thermally excited capillary waves. The quantity in square brackets leads to a correction term that is definitely subleading for large wave numbers, $q \gg \Lambda$, where $\Lambda = \xi_{\parallel}^{-1}$ is an upper cutoff.

## 7  Conclusions

In this paper we applied a field-theoretical approach to describe, in a completely novel way, correlations in phase separating systems in the presence of boundaries encompassing the relevant regimes of entropic repulsion and wetting in two dimensions. Our results can be summarized as follows.

The regime of entropic repulsion due to a hard wall is studied by fixing the extremities of the interface in two points separated by a distance $R$. Field theory yields exact results for order parameter profiles and correlations in the regime of subcritical temperatures with $R \gg \xi_{\mathrm{b}}$. The theory predicts that the order parameter is affected by a finite-size correction of order $R^{-1/2}$ which is proportional to the passage probability density of a Brownian excursion, $P(x, y)$. We have tested this prediction in two different but complementary approaches. Firstly, we measured the order parameter profile and showed that upon including the correction term $\mathscr{A}P(x, y = 0)$ an excellent agreement with MC simulations is obtained. This comparison, which is carried out in the absence of free parameters, resolves a longstanding discrepancy observed in early simulations [42] in which the order parameter profile obtained in simulations has been compared against the leading order profile which, in our notations, is proportional to $\Upsilon(\eta)$. Secondly, we have extracted the probability density $P(x, y = 0)$ directly by sampling interfacial crossings on the horizontal axis. For this analysis we properly defined the *off-critical* interface both on the medial lattice and in terms of the loop erased random walk. The prediction for $P(x, 0)$ is successfully tested without any fitting parameter. Then, we also compared MC simulations with the analytical prediction for the connected two-point function of the order parameter. In particular, we showed that correlations are long-range along the interface and are confined within the interfacial region for the regime of entropic repulsion.

In the second part of the paper we examined one- and two-point correlations at the wetting transition. In particular, we examined both the energy density and the order parameter field. We argue that $n$-point energy density correlations (at wetting) are proportional to a $n$-point passage probability density; in this paper we considered $n = 1$ and $n = 2$ but the result should apply to arbitrary $n$, in analogy with the case of the strip [29]. On the basis of a probabilistic interpretation we derived both one- and two-point correlation functions of the order parameter field. The probabilistic reconstruction of one-point functions, which is encoded in the identities (2) and (3), has been established in this paper. The extension to the case of two-point functions, encoded in (6) and (7), will be reported in full detail in a forthcoming publication [89]. The result we obtain for the energy-density correlation coincides with the existing result obtained for the Ising model [88] and for Solid-On-Solid models [58]. However, the approach we employed applies to a wide class of models which exhibit the wetting phenomenology, meaning the existence of a bound state in the boundary $S$-matrix. What we showed in this paper is that such a singularity exhibited by matrix elements in field theory is the counterpart of the so-called wetting pole in the Ising model [88]. In analogy with the regime of entropic repulsion, correlations are still long ranged along the interface but for the wetting regime the characteristic entropic factor $\propto x^2$ does not occur; see (16) and (23), which apply for $y \ll \Lambda^{-1}$ with $\Lambda^{(\mathrm{er})} = R^{-1}$ and $\Lambda^{(\mathrm{w})} = \xi_{\parallel}^{-1}$. Moreover, we show that both the regimes

share similarities when a suitable identification of the associated parallel and perpendicular correlation lengths is performed; see the last paragraph of Sec. 5.3.

We illustrated how the classical notion of ISF can be extended to system defined in the semi-space bounded by a flat wall. For both the regimes of entropic repulsion and wetting the large-$q$ asymptotic form of the interface structure factor coincides with the familiar result from capillary wave theory, i.e., $S(q) \sim A/q^2$. In both the regimes we find the same prefactor $A = (\rho_l - \rho_v)^2/\gamma_{lv}$. This large-$q$ result is obtained for $q \gg \Lambda$ with a cutoff $\Lambda$ that depends on the regime of interest; $\Lambda^{(er)}$ for entropic repulsion and $\Lambda^{(w)}$ at wetting. We also identify the next-to-leading term in the large-$q$ expansion and show that it carries additional powers in $q$ and is a distinctive feature of the regime.

We stress that in our calculation of the ISF we do not rely on a particular definition of the interface since $S(q)$ is defined as a twofold integral of a connected density-density correlation function, and the latter does not rely on any definition of the interface location. However, in order to make the bridge between field theory and effective theories such as the capillary wave model, we prove the equivalence between the calculation based on the density-density correlation function and on the height-height correlation function, which indeed assumes a specific location of the interface. This analysis is carried out for both entropic repulsion and wetting.

Finally, we described in a pedagogical fashion the essential ideas underlying the exact field theory of phase separation [17–20, 90, 91]. The material covered in Sec. 2 is certainly not an exhaustive review on the subject but rather the intent is to show in a rather simple way how to establish a clear connection between the the language of field theory and the one of interfacial phenomena.

Looking at perspectives, it would be interesting to further investigate how the interplay between geometry and universality classes affects the interface structure factor. The promising candidate for these studies is the $q$-state Potts model in a wedge-shaped geometry [20]. For such a model it is possible to characterize both the filling transition [92, 93] and the elusive symmetry named *wedge covariance* [48–50, 94] at the level of correlation functions. In those cases it is expected to find a non-trivial geometry-dependent interface structure coefficient $\mathscr{A}$. The results presented in this paper show how to initiate these promising studies. Other interesting directions point towards three-dimensional systems with the recently establishes techniques of Refs. [95, 96].

## Acknowledgements

A. S. is grateful to Gesualdo Delfino for collaborations on closely related topics, Douglas B. Abraham for inspiring discussions and for having drawn the attention to the numerical results of Ref. [42], Robert Evans for discussions and S. Dietrich for helpful correspondence.

**Funding information** A. S. acknowledges FWF Der Wissenschaftsfonds for funding through the Lise-Meitner Fellowship (Grant No. M 3300-N).

## A One-point functions at wetting

As a warmup, and to fix the notations, we con calculate the partition function for the system depicted in Fig. 1,

$$Z = {}_{B_b}\langle 0|\mu_{ba}(R/2)\mu_{ab}(-R/2)|0\rangle_{B_b}, \tag{A.1}$$

the notation $|0\rangle_{B_b}$ stands for the vacuum state in which the wall has fixed boundary condition with spin in state $b$. Then, $\mu_{ab}(\pm R/2)$ is the boundary condition changing operator which implements the switch of boundary condition from $b$ to $a$ at the pinning point $(0, \pm R/2)$. The calculation is performed by inserting a resolution of the identity between the $\mu$-operators and then by summing over fundamental excitations of the bulk theory [18,20]. Excitations in two dimensions are topological kink particles $|K_{ab}(\theta_j)\rangle$ whose worldline propagation in Euclidean space-time corresponds to the interface between coexisting phases. The rapidity variable $\theta_j$ parametrizes energy and momentum satisfying the relativistic dispersion relation $e_j^2 - p_j^2 = m^2$ through $e_j = m \cosh \theta_j$, $p_j = m \sinh \theta_j$. The matrix element of $\mu_{ba}$ between the vacuum and the single-kink state $|K_{ab}(\theta_j)\rangle$ reads

$$_{B_b}\langle 0|\mu_{ba}(y)|K_{ab}(\theta)\rangle = e^{-my}\mathcal{F}_\mu(\theta). \tag{A.2}$$

$m$ is the kink mass and $\mathcal{F}_\mu(\theta)$ is the form factor of the boundary condition changing operator $\mu_{ba}$; these quantities have been studied intensively in the framework of massive integrable quantum field theories and are known for a wide class of models [97–99]. The fact that $\mu_{ba}$ exhibits a non-vanishing matrix element with the one-kink state corresponds to the pinning of a *single domain wall* in those points in which the boundary condition switches from $a$ to $b$ [18,20]. As a side remark, we mention that in certain models $\mathcal{F}_\mu(\theta) = 0$ and the first nontrivial form factor is a two-particle one [21]. In these cases the pinning of a double kink corresponds to the formation of an intermediate phase which ultimately leads to a different phase separation pattern [21]. The overall exponential in (A.2) follows by using translational invariance of bulk fields,

$$\Phi(x, y) = e^{ixP + yH}\Phi(0, 0)e^{-ixP - yH}, \tag{A.3}$$

where $H$ and $P$ are the Hamiltonian and momentum operators in field theory, i.e.,

$$H|K_{ab}(\theta)\rangle = m \cosh \theta |K_{ab}(\theta)\rangle, \tag{A.4}$$

$$P|K_{ab}(\theta)\rangle = m \sinh \theta |K_{ab}(\theta)\rangle. \tag{A.5}$$

The regime $R/\xi_b \gg 1$ – which is the one pertinent for the study of phase separation – is completely captured by the low-energy behavior of form factors, i.e., $\theta \to 0$. The asymptotic form of the partition function is dominated by the single-particle term in the spectral series and therefore

$$Z \simeq \int_0^\infty \frac{d\theta}{2\pi}|\mathcal{F}_\mu(\theta)|^2 e^{-mR\cosh\theta}, \tag{A.6}$$

where $\simeq$ stands for the omission of subleading contributions involving the propagation of more than one-particles. The wall restricts the integration to positive rapidities of incoming kinks which are elastically reflected by the wall upon collision. In the case of entropic repulsion the boundary form factor admits the low-energy behavior

$$\mathcal{F}_\mu(\theta) = i\mathfrak{a}\theta + \mathfrak{b}\theta^2 + \mathcal{O}(\theta^3). \tag{A.7}$$

The model-dependent coefficients $\mathfrak{a}$ and $\mathfrak{b}$ – and all subsequent ones in the expansion – are known for integrable quantum field theories with boundaries [97–100]. We refer the reader interested in integrable field theories to [101,102] and [87] for the bulk and boundary cases, respectively. The linear behavior at low rapidities exhibited by $\mathcal{F}_\mu(\theta)$ is ultimately responsible for the entropic repulsion of the interface from the wall [18]. In particular, no singular behavior is expected in the physical strip $\text{Im}(\theta) \in (0, \pi)$. In contrast to this situation, in the wetting regime the scattering of a particle off the boundary exhibits a simple pole in the boundary $S$-matrix, which behaves as $R(\theta) \sim ig^2/(\theta - iu)$ for $\theta \to iu$ [87]. The pole is interpreted as a bound state in the particle+wall system, as illustrated in (A.9). It follows that the matrix

element $\mathcal{F}_\mu(\theta)$ inherits a pole in the physical strip for a virtual value of the rapidity $\theta = iu$. The singular behavior in the closeness of the pole is of the form

$$\mathcal{F}_\mu(\theta) \sim \frac{ig}{\theta - iu}\langle 0|\mu|B'_a\rangle, \qquad \theta \to iu, \tag{A.8}$$

where $g$ is the strength of the boundary bound state, the latter is proportional to the residue of the boundary form factor at the virtual rapidity $\theta = iu$. Equation (A.8) is pictorially represented in (A.9). The wall adsorbs phase $b$ as a result of the bound state pole in the emission amplitude of the kink $K_{ab}$.

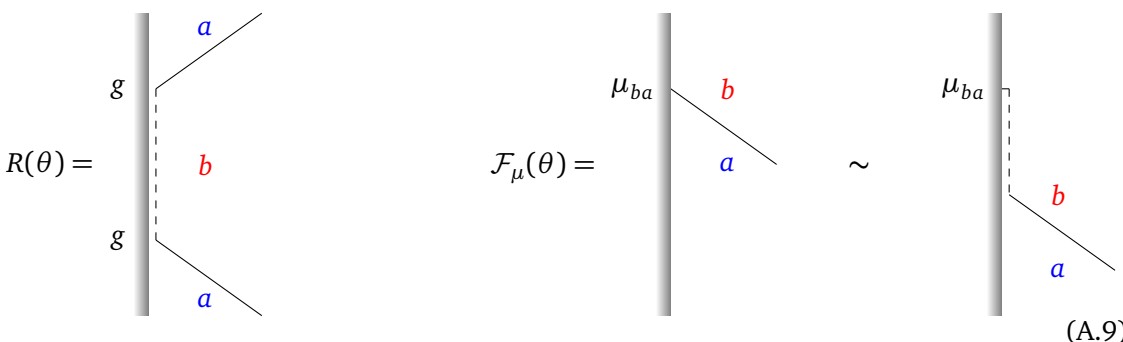

$$\tag{A.9}$$

The large-$mR$ asymptotic behavior of the partition function is dominated by the wetting pole, which yields

$$Z_{\mathrm{w}} \simeq \frac{g^2}{2u}e^{-mR\cos u}. \tag{A.10}$$

This implies that the excess free energy associated to the interface is

$$\gamma_{wa} - \gamma_{wb} = -\lim_{R\to\infty}\frac{1}{R}\ln Z. \tag{A.11}$$

Some comments are in order. Let us consider the wetting regime, i.e., $Z = Z_{\mathrm{w}}$. By plugging the partition function (A.10) into (A.11), we obtain $\gamma_{wa} - \gamma_{wb} = m\cos u$. The latter is exactly *Young's equation* [47]; $m = \gamma_{ab}$ can be identified with the surface tension of the, say, liquid-vapour interface, and $u$ is the associated contact angle. In the regime dominated by entropic repulsion the partition function is $Z_{\mathrm{er}} \sim R^{-3/2}\exp(-mR)$ [18]. In this case (A.11) yields $\gamma_{wa} - \gamma_{wb} = \gamma_{ab}$, which is the so-called *Antonov's rule*[12] [104]. This result is consistent with a contact angle that formally vanishes, i.e., $u = 0$. Of course, the contact angle at the pinning point is not zero but nonetheless this result can be intuitively understood since the droplet – within the limit $R \to \infty$ – undergoes macroscopic fluctuations which completely wet the wall. We refer to [47] for an accurate account on the domain of validity of Antonov's rule.

Let us consider a field $\Phi(x,y)$ that can be either the energy density or the spin field. Following the method outlined [18], the connected part of the one-point function is

$$\langle\Phi(x,y)\rangle^{\mathrm{CP}}_{ab} \simeq \frac{1}{Z}\int_\mathcal{C}\frac{d\theta_1}{2\pi}\int_\mathcal{C}\frac{d\theta_2}{2\pi}\mathcal{F}_\mu(\theta_1)\mathcal{F}^*_\mu(\theta_2)F_\Phi(\theta_{12}+i\pi)Y(\theta_1,\theta_2), \tag{A.12}$$

where

$$Y(\theta_1,\theta_2) = \exp\left[-m\left(\frac{R}{2}-y\right)\cosh\theta_1 - m\left(\frac{R}{2}+y\right)\cosh\theta_2 + imx(\sinh\theta_1 - \sinh\theta_2)\right], \tag{A.13}$$

and $F_\Phi(\theta_{12}) = \langle 0|\Phi(0,0)|K_{ab}(\theta_1)K_{ba}(\theta_2)\rangle$ is the two-particle form factor of the operator $\Phi$ [102]. The integration contour $\mathcal{C}$ can be split into a path which runs on the real axis and a circular loop around the bound state pole, as prescribed in [87]; see Fig. 11.

---

[12]See [103] for mean-field studies of Antonov's rule applied to line tensions.

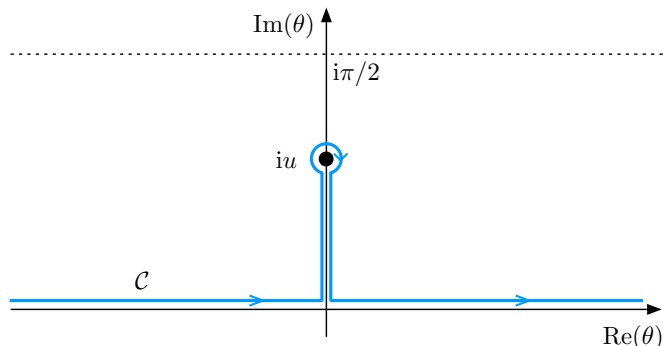

Figure 11: The integration contour $\mathcal{C}$ surrounding the wetting pole $\theta = \mathrm{i}u$ [87].

In the regime of entropic repulsion the boundary form factor $\mathcal{F}_\mu(\theta)$ admits the low-energy expansion (A.7). The form factor $F_\sigma(\theta + \mathrm{i}\pi)$ admits the following low-energy expansion [17, 105]

$$F_\sigma(\theta + \mathrm{i}\pi) = \frac{\mathrm{i}(\langle\sigma\rangle_a - \langle\sigma\rangle_b)}{\theta} + c_0 + c_1\theta + \dots, \tag{A.14}$$

the kinematical pole $1/\theta$ yields the leading-order term in the large-$R$ expansion of the order parameter profile. The terms $c_k$ with $k \geqslant 0$ contribute to interface structure corrections. In particular, $c_0$ is the coefficient entering the interface structure amplitude $\mathscr{A}$ where

$$\mathscr{A} = \frac{c_0}{m} + \frac{\mathfrak{b}}{\mathfrak{a}}\frac{\Delta\langle\sigma\rangle}{m}. \tag{A.15}$$

$\Delta\langle\sigma\rangle = \langle\sigma\rangle_a - \langle\sigma\rangle_b$ is the jump of order parameter across the $a - b$ interface and $m$ is the surface tension [28]. In general, we can decompose (A.15) into the sum of two terms by writing $\mathscr{A} = \mathscr{A}^{\text{half-plane}}$ with

$$\mathscr{A}^{(\text{half-plane})} = \mathscr{A}^{(\text{strip})} + \delta\mathscr{A}, \tag{A.16}$$

where

$$\mathscr{A}^{(\text{strip})} = \frac{c_0}{m}, \tag{A.17}$$

is the result corresponding to the strip geometry [17], and

$$\delta\mathscr{A} = \frac{\mathfrak{b}}{\mathfrak{a}}\frac{\Delta\langle\sigma\rangle}{m}, \tag{A.18}$$

is an excess contribution distinctive of the half-plane geometry. The term proportional to $\mathfrak{b}/\mathfrak{a}$ is a specificity of the half-plane geometry which does not occur for phase separation in the strip [17]. In contrast with the case of the Ising model on the strip – in which $c_0 = 0$ rules out the correction $\propto R^{-1/2}$ – the term $\mathfrak{b}/\mathfrak{a}$ does not vanish on the half-plane and gives $\mathscr{A}^{(\text{half-plane})} = M/m$.

Let us consider now the wetting regime characterized the existence of the wetting pole in $\theta = \mathrm{i}u$. The calculation of the density profile follows from (A.12). The vertical sides of the path $\mathcal{C}$ which join the real axis with the pole give a vanishing contribution since the integrand does not exhibits branch cuts. As a matter of fact the $\mathcal{C}$-integral is dominated by the pole contributions originated by the boundary form factors $\mathcal{F}_\mu(\theta_1)$ and $\mathcal{F}_\mu^*(\theta_2)$. Taking into account that such residues are picked up by loops with opposite orientations, we find

$$\langle\Phi(x,y)\rangle_{ab}^{\text{CP}} \simeq \frac{g^2}{Z} F_\Phi(2\mathrm{i}u + \mathrm{i}\pi) Y(\mathrm{i}u, -\mathrm{i}u), \tag{A.19}$$

where CP stands for the contribution generated by the connected matrix element, which is depicted in (A.20) as a blob connecting an incoming and outgoing momenta $\theta_1$ and $\theta_2$. Within the pictorial representation, the boundary bound state detaches from the wall with contact angle $\Theta = u$, reaches the spin field, and then it is reabsorbed on the wall.

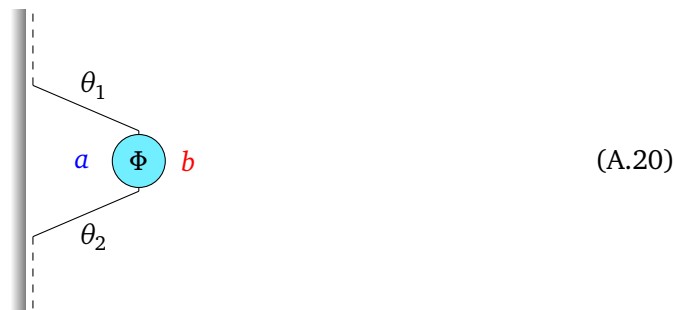

$$(A.20)$$

We consider now the energy density profile, $\Psi = \varepsilon$. Since the case we examine is $u \to 0$ the above yields

$$\langle \varepsilon(x, y) \rangle_{ab}^{\text{CP}} \simeq 2u F_\varepsilon(\mathrm{i}\pi) \mathrm{e}^{-2mux} . \tag{A.21}$$

The overall constant $F_\varepsilon(\mathrm{i}\pi) \sim m^{X_\varepsilon}$, where $X_\varepsilon$ is the scaling dimension of the field $\varepsilon$ [106]. We observe that now the profile is translationally invariant along the wall ($y$-direction). Then, the natural length scale associated to the exponential attenuation is the perpendicular correlation length $\xi_\perp$ defined by

$$\xi_\perp = (mu)^{-1} . \tag{A.22}$$

Since $u \sim T_w - T$, it follows that $\xi_\perp \sim (T_w - T)^{-\nu_\perp}$ with the critical exponent $\nu_\perp = 1$, as reported in Sec. 5.2. We can repeat the above analysis for the order parameter profile. Now we have

$$\langle \sigma(x, y) \rangle_{ab}^{\text{CP}} \simeq 2u F_\sigma(2\mathrm{i}u + \mathrm{i}\pi) \mathrm{e}^{-2mux} . \tag{A.23}$$

The expansion (A.14) implies that $u F_\sigma(2\mathrm{i}u + \mathrm{i}\pi)$ for $u \to 0$ yields the jump of order parameter across the interface [105], i.e.,

$$\lim_{u \to 0} 2u F_\sigma(2\mathrm{i}u + \mathrm{i}\pi) = \langle \sigma \rangle_a - \langle \sigma \rangle_b . \tag{A.24}$$

By imposing the appropriate boundary condition deep into the bulk phase, i.e.,

$$\langle \sigma(x \to +\infty, y) \rangle_{ab} = \langle \sigma \rangle_b , \tag{A.25}$$

we obtain

$$\langle \sigma(x, y) \rangle_{ab} = \langle \sigma \rangle_b + (\langle \sigma \rangle_a - \langle \sigma \rangle_b) \mathrm{e}^{-2mux} , \tag{A.26}$$

which is valid for large $x$. Parenthetically, we observe that the offset term $\langle \sigma \rangle_b$ can be obtained by computing the disconnected part of the matrix element. The latter can be depicted as a strand connecting the rapidities $\theta_1$ and $\theta_2$ with the field $\sigma$ surrounded by the in the $b$-phase.

The energy density and order parameter profiles, respectively (A.21) and (A.26), are consistent with a probabilistic interpretation in which interfacial fluctuations are distributed according to the probability density

$$P^{(\text{w})}(x) = (2/\xi_\perp) \mathrm{e}^{-2x/\xi_\perp} , \tag{A.27}$$

and the order parameter profile is thus reconstructed by averaging – with the passage probability $P^{(\text{w})}$ – interfacial crossings along the $x$-axis, i.e.,

$$\langle \sigma(x) \rangle_{ab} = \int_0^\infty \mathrm{d}u \, P^{(\text{w})}(u) \sigma_{ab}(x|u) , \tag{A.28}$$

with the conditional, or sharp, magnetization profile $\sigma_{ab}(x|u) = \langle \sigma \rangle_a \theta(x - u) + \langle \sigma \rangle_b \theta(u - x)$.

# B   Two-point functions at wetting

By following the guidelines outlined in Appendix A it is possible to compute the pair correlation functions at wetting. The calculation of the order parameter pair correlation turns out to be rather cumbersome because it involves two kinematic poles stemming from spin fields and two bound state poles stemming from boundary form factors; see (A.14) and (A.7), respectively. The calculation of the energy density profile is however much simpler as the kinematic poles do not arise.

In order to provide a glimpse on the calculation, we only illustrate the relevant diagrams which need to be taken into account. Considering the energy density field, the diagrams to be computed are those depicted in (B.1).

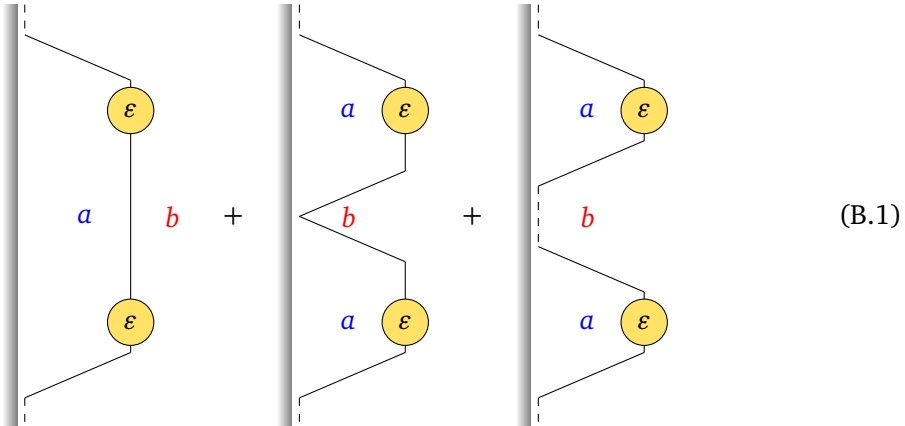

$$(B.1)$$

The first and the second diagrams are similar to those already encountered for the case of entropic repulsion [28]. Here, however, the incoming and outgoing momenta are those of the *boundary bound state* (dashed lines). In the second diagram the intermediate particle between the two operators is scattered by the wall and acquires a boundary $S$-matrix $R(\theta)$ [87]. In the last diagram the interaction with the wall occurs via a boundary bound state. The way to tackle these diagram retraces the analysis presented in a recent preprint that doesn't need to be reproduced here, we therefore refer the interested reader to [28] for a detailed account of the technicalities. The result we obtain for the connected energy density is

$$\langle \varepsilon(x_1, y_1) \varepsilon(x_2, y_2) \rangle^c_{-+} \propto \xi_\perp^{-2} \Pi_2(X, \bar{X}; Y), \tag{B.2}$$

where $X = X_1 - X_2, \overline{X} = X_1 + X_2$ are rescaled coordinates appearing in (20) and

$$\Pi_2(X, \bar{X}; Y) = (\pi Y)^{-1/2} e^{-\overline{X} - Y} \left( e^{-X^2/4Y} + e^{-\overline{X}^2/4Y} \right) + 2 e^{-2\overline{X}} \mathrm{erfc}\left( \frac{\overline{X}}{2\sqrt{Y}} - \sqrt{Y} \right). \tag{B.3}$$

The probability density (21) and the rescaled version (B.3) are normalized, meaning that

$$\int_0^\infty dx_1 \int_0^\infty dx_2 \, P_2^{(w)}(x_1, x_2; y_1 - y_2) = 1, \tag{B.4}$$

and their marginal retrieves the one-body probability density, i.e.,

$$\int_0^\infty dx_2 \, P_2^{(w)}(x_1, x_2; y_1 - y_2) = P^{(w)}(x_1). \tag{B.5}$$

The latter does not depend on $y_1$ and $y_2$ because of translational invariance.

We can consider now the density-density correlation function. By using the probabilistic representation given in the main body of the paper, i.e.,

$$\langle\sigma(x_1,y_1)\sigma(x_2,y_2)\rangle_{-+} = \int_0^\infty \mathrm{d}u_1 \int_0^\infty \mathrm{d}u_2\, P_2^{(\mathrm{w})}(u_1,u_2;y_1-y_2)\sigma_{-+}(x_1|u_1)\sigma_{-+}(x_2|u_2). \tag{B.6}$$

For $y_1 > y_2$ the calculation entails

$$\langle\sigma(x_1,y_1)\sigma(x_2,y_2)\rangle_{-+} = M^2 \mathcal{G}(X_1,X_2;Y_1-Y_2), \tag{B.7}$$

the scaling function $\mathcal{G}$

$$\mathcal{G}(X_1,X_2;Y) = -1 + 2\mathrm{e}^{-2(X_1+X_2)}(1-\mathcal{K}_{+,-}) + 2\mathrm{e}^{-2X_1}\mathcal{K}_{-,-} - 2\mathrm{e}^{-2X_2}\mathcal{K}_{-,+} + 2\mathcal{K}_{+,+}, \tag{B.8}$$

is expressed in terms of the compact notation

$$\mathcal{K}_{\alpha,\beta}(X_1,X_2;Y) \equiv \mathrm{erf}\left(\frac{X_1+\alpha X_2}{2\sqrt{Y}} + \beta\sqrt{Y}\right), \qquad \alpha,\beta = \pm 1. \tag{B.9}$$

An analogous scaling form is obtained for the connected correlation function,

$$\begin{aligned}
\langle\sigma(x_1,y_1)\sigma(x_2,y_2)\rangle_{-+}^{\mathrm{c}} &= \int_0^\infty \mathrm{d}u_1 \int_0^\infty \mathrm{d}u_2\left[ P_2^{(\mathrm{w})}(u_1,u_2;y_1-y_2) - P^{(\mathrm{w})}(u_1)P^{(\mathrm{w})}(u_2)\right] \\
&\qquad\qquad \times \sigma_{-+}(x_1|u_1)\sigma_{-+}(x_2|u_2) \\
&= M^2 \mathcal{G}^{\mathrm{c}}(X_1,X_2;Y_1-Y_2),
\end{aligned} \tag{B.10}$$

where

$$\mathcal{G}^{\mathrm{c}}(X_1,X_2;Y) = \mathcal{G}(X_1,X_2;Y) - \left(1-2\mathrm{e}^{-2X_1}\right)\left(1-2\mathrm{e}^{-2X_2}\right). \tag{B.11}$$

## C  Interfacial fluctuations and structure factor

For an infinitely long domain wall the interface structure factor is

$$S(q) = \frac{1}{2}\int_{-\infty}^{+\infty} \mathrm{d}y\, \mathrm{e}^{\mathrm{i}qy} \int_0^\infty \mathrm{d}x_1 \int_0^\infty \mathrm{d}x_2 \langle\sigma(x_1,|y|)\sigma(x_2,-|y|)\rangle_{-+}^{\mathrm{c}}. \tag{C.1}$$

The integrated density-density correlation function can be calculated out as follows

$$\begin{aligned}
\int_0^\infty \mathrm{d}x_1 \int_0^\infty \mathrm{d}x_2 \langle\sigma(x_1,y_1)\sigma(x_2,y_2)\rangle_{-+}^{\mathrm{c}} &= M^2\xi_\perp^2 \int_0^\infty \mathrm{d}X_1 \int_0^\infty \mathrm{d}X_2\, \mathcal{G}^{\mathrm{c}}(X_1,X_2;Y_1-Y_2) \\
&= M^2\xi_\perp^2 \mathcal{I}(Y_1-Y_2),
\end{aligned} \tag{C.2}$$

where $\mathcal{I}(Y)$ is the function given in (34). On the other hand, also the connected height-height correlation function can be expressed in terms of the function $\mathcal{I}(Y)$:

$$\begin{aligned}
\mathscr{C}_{\mathrm{wet}}(y_1-y_2) &= \overline{h(y_1)h(y_2)} - \overline{h(y_1)}\,\overline{h(y_2)} \\
&= \int_0^\infty \mathrm{d}x_1 \int_0^\infty \mathrm{d}x_2\, x_1 x_2\left[ P_2^{(\mathrm{w})}(u_1,u_2;y_1-y_2) - P^{(\mathrm{w})}(x_1)P^{(\mathrm{w})}(x_2)\right] \\
&= \frac{\xi_\perp^2}{4}\mathcal{I}((y_1-y_2)/\xi_\parallel).
\end{aligned} \tag{C.3}$$

The above implies that $S(q)$ can be cast in the following way

$$S(q) = \frac{1}{4} M^2 \xi_\perp^2 \xi_\parallel \int_{-\infty}^{+\infty} dY \, e^{iQY} \mathcal{I}(|Y|), \tag{C.4}$$

where $Q = q\xi_\parallel/2$ is the rescaled wavenumber. The Fourier transform of $\mathcal{I}(|Y|)$ can be parametrized in a form that is suitable for successive considerations

$$\int_{-\infty}^{+\infty} dY \, e^{iQY} \mathcal{I}(|Y|) \equiv \frac{8}{Q^2} \Xi(Q), \tag{C.5}$$

where $\Xi(Q)$ is the function

$$\Xi(Q) = 1 + \frac{1}{Q} \left( \sqrt{Q^2+1} - Q - 1 \right) \sqrt{\sqrt{Q^2+1} + Q}. \tag{C.6}$$

The nested radicals in (C.6) can be further simplified and eventually it can be brought in the simpler form

$$\Xi(Q) = 1 - \left( \frac{1}{2} \left( \sqrt{Q^2+1} + 1 \right) \right)^{-1/2}, \tag{C.7}$$

therefore

$$S(q) = \frac{8M^2 \xi_\perp^2}{q^2 \xi_\parallel} \Xi(Q). \tag{C.8}$$

The rest of the calculation is trivial algebra. We bring in the jump of density across the interface, i.e., we write $(2M)^2 = (\rho_l - \rho_v)^2$, and obtain

$$S(q) = \frac{2\xi_\perp^2}{\xi_\parallel} \frac{(\rho_l - \rho_v)^2}{q^2} \Xi(Q). \tag{C.9}$$

Although both $\xi_\perp$ and $\xi_\parallel$ diverge upon approaching wetting, the combination appearing in (C.9) stays finite because $\xi_\parallel = 1/mu$ and $\xi_\perp = 2/mu^2$ jointly with $m = \gamma_{lv}$ yield the identity

$$\frac{\xi_\parallel}{2\xi_\perp^2} = \gamma_{lv}. \tag{C.10}$$

The above implies that the overall coefficient multiplying the $1/q^2$ dependence in (C.11) is the standard one expected for the interface structure factor, i.e.,

$$S(q) = \frac{(\rho_l - \rho_v)^2}{\gamma_{lv} q^2} \Xi(q\xi_\parallel/2), \tag{C.11}$$

which is the result (35) quoted in the main body of the paper.

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
