# Peer review of "Interfacially adsorbed bubbles determine the shape of droplets"

_SciPost Physics, doi:SciPost Phys. 15, 164 (2023)_

## Round 1 · Referee Report · Anonymous (Referee 1) · 2023-5-31

Strengths

The work presents interesting and important results in the field of near-critical interfacial phenomena.

Report

The authors consider the two-dimensional Ising model on the half-plane x > 0 with boundary conditions enforcing a droplet shape. They consider two different scenarios for the droplet shape: entropic repulsion where the droplet is partially covering the absorbing surface, and the wetting transition where the droplet is covering the complete surface. For both scenarios they use the equivalence between the near-critical Ising model and a relativistic quantum field theory to obtain expressions for the density profile (Eqs. (1) and (7)) and correlation functions (Eqs. (5) and (11)). For the former scenario the analytical results are compared with MC simulations. Finally, in Sec. (5) the authors provide a technique to calculate the interface structure factor for semi-confined systems.

The work presents interesting and important results in the field of near-critical interfacial phenomena. However, some issues must be addressed and clarified before the paper can be accepted; please see my remarks below.

Specific remarks:

  • Introduction: ”From the theoretical side, boundary-induced effects on near-critical systems have been intensively studied by means of several techniques ranging from mean field theory, perturbative field theory [9–11], and numerical simulations [12].”(p2)

References are missing for “mean field theory”

1) Introduction: - ”...the exact analytic form of correlations in the presence of strongly fluctuating interfaces is largely unknown...” (p3). This statement seems to contradict the result of Onsager mentioned on the top of page 3:

  • ” The celebrated Onsager’s solution yields the exact result for the decay of scaled truncated two-point function, … [23]” (p3). The authors probably mean that the exact analytic form of correlations in the presence of a boundary is largely unknown. If this is so, the authors should make this statement more precise.

2) Entropic repulsion - ”For the Ising model with a = − and b = +1 this protocol introduces a droplet of negative magnetization enclosed in the Peierls contour of Fig. 1 (a) [47].” (p5)

I suggest to briefly explain what a Peierls contour is. Moreover, a = − should be written as a = −1.

  • ”The mapping between variables in the lattice gas and Ising model is n i = (1 + s i )/2, where n i ∈ 0, 1 stands for an empty/filled site [6,24,48]. (p5)

An explanation/elaboration of s_i is seemingly missing. It is obvious that these are the spin variables s_i = ±1, but this does not seem to be stated anywhere.

  • The Ising Hamiltonian is never defined. Although many readers will be familiar with the Ising Hamiltonian, for completeness it should at least be stated once. This way it will also be immediately clear how the coupling J and J_b enter the Hamiltonian.

  • The authors assign the variable $a$ to two different quantities. First they write ”For the Ising model with a = − ...” (p5) and shortly after ”...and those in the adjacent layer is taken to be of the form J_b = aJ, where 0 ≤ a ≤ 1, meaning that J_b is a weakened bond;” (p5)

This is confusing. It seems that the first a = −1 does not have to be introduced if the authors introduce the spin variables s_i = ±1.

  • ”... there exist a wetting temperature T w (a) < T c such that T w (a → 0 + ) = T c − and ...” (p5) Here both T_c and T^-_c are not defined. Most likely T_c is the critical temperature (which the referee will henceforth assume), but this should be stated explicitly.

  • ” To be definite, in our simulations for the case of entropic repulsion we simply take J_b = J, corresponding to a = 1, meaning that for all subcritical temperatures the system is non-wet.” Should subcritical temperatures not be positive temperatures, since T_w (a \to 1^− ) = 0^+ ?

  • ”... for the Ising model ⟨σ⟩ = ±M where M \sim (T_c − T )^{1/8} ...” (p6) To be consistent with previous notation, and to avoid the need for introducing useless variables, they should change ⟨σ⟩ to ⟨s⟩ or change the s i on page 5 to σ_i . Furthermore, the relation M \sim (T_c − T )^{1/8} only applies close to the critical temperature. It would be more correct if the authors state the exact result M = [1 − \sinh^{−4}(2J/k_B T)]^{1/8}

  • One of the main results of this manuscript is shown in Eq. (1). Since the authors always write scaling relations close to the critical temperature, i.e. M \sim (T_c − T )^{1/8} and \xi_b \sim (T_c − T )^{-\nu} , it is not clear whether Eq. (1) only applies close to the critical temperature T_c , or for arbitrary temperatures. If the former is the case (which is suggested according to Sec. 6) then this must be explicitly stated. If the latter is the case, then the fully general result for M and \xi_b in terms of the coupling J and temperature T should be stated accordingly.

  • ”The correction term \mathcal{A}P (x, y)\propto R^{−1/2} , which we have identified, turns out to resolve the aforementioned mismatch and eventually yields – if included in the profile – an accurate comparison between theory and numerics, as shown in Fig. 2. The agreement is perfect due to such a term and without taking it into account the MC data fall systematically away from the analytic prediction.” (p7)

To be able to faithfully make such a statement the authors must include the analytical profile without the correction terms in Fig. 2. Without this the reader simply cannot see the importance of the correction term, since it can very well be that without the correction term the density profiles already match quite nicely.

  • ”In general, it is possible to regard the interface as the result of an exploration process which starts from one pinning point and ends at the other; see points denoted “in” and “out” in Fig. 3.” (p8)

The idea of regarding the interface as a Brownian bridge is not new and has been pursued in quite a few previous works. See for example:

  • M. E. Fisher, Walks, walls, wetting, and melting, J. Stat. Phys. 34, 667 (1984).
  • Hryniv, Ostap. On local behaviour of the phase separation line in the 2D Ising model. Probability theory and related fields. 110.1 (1998): 91-107.
  • Ganguly, Shirshendu, and Reza Gheissari, Local and global geometry of the 2D Ising interface in critical prewetting. (2021): 2076-2140.
  • K. Blom, N. Ziethen, D. Zwicker, A. Godec, Thermodynamically consistent phase-field theory including nearest-neighbor pair correlations, Phys. Rev. Res. 5, 1 (2023).

It therefore seems to be appropriate to cite these works in the above mentioned sentence.

  • Figures 2, 4, and 5 show a comparison between the analytical theory and MC simulations. However, many details about the MC simulations are missing. For example, the number of trajectories that were considered, the uncertainty of the data points, and the conversion from the discrete lattice spacing to a continuous length variable x. These details must be mentioned/stated in the manuscript.

3) Wetting transition

  • The variable m is introduced for the first time on page 11, whereas its explanation only comes much later on page 12 under equation (9): ” … where m is the surface tension of the interface”. This explanation should come directly after the first mentioning of m on page 11.

  • In Eq. (8) we find P_2 has 3 variables, whereas in Eq. (10) we find it has 4. This is inconsistent and needs to be resolved.

  • Compared with the previous section the referee was wondering why Eqs. (7) and (11) – which provide analytical results for the density profile and correlation function – are not compared with MC simulations (given that these were carried out)? For completeness and scientific correctness, the authors should provide a comparison with MC simulations here. Considering the fact that they have done this for the entropic repulsion case, this should not be too difficult.

4) Theoretical framework in a nutshell

  • This section is very well-written and provides a clear analogy between the relativistic quantum field theory and the interface theory. In fact, this section (or a “diluted” version thereof) would suit much better at the beginning of the manuscript before the section Entropic repulsion. This will also immediately resolve an issue mentioned above, since this section clearly shows that the applied techniques are only valid close to the critical temperature T_c.

5) Conclusion:

  • ”Field theory yields exact results for order parameter profiles and correlations in the regime of subcritical temperatures with R >> ξ b .” (p19)

This is confusing w.r.t. to following statement in Sec. 6:

”The near-critical behavior in 2D can be described by analytic continuation of a (1 + 1)- relativistic quantum field theory to a 2D Euclidean field theory in the plane (y = −it).” (p15)

So, do the results apply to near-critical or to sub-critical temperatures? This needs to be clarified.

Typos: - The analytical progress along mentioned \to The analytical progress mentioned - page 6: and P(x; y) are super-universal, \to and P (x, y) are super-universal - page 14: and yields the solid black arc of ellipse shown in Fig. 1 (c)\to and yields the solid black arc of ellipse shown in Fig. 1 (b) - page 20: The results presented in this paper and show how to \to The results presented in this paper show how to ...

  • validity: high
  • significance: good
  • originality: good
  • clarity: good
  • formatting: excellent
  • grammar: good

Author:  Alessio Squarcini  on 2023-07-14  [id 3810]

(in reply to Report 1 on 2023-05-31)

REPLY TO REPORT 1

REFEREE’S COMMENT: The authors consider the two-dimensional Ising model on the half-plane x > 0 with boundary conditions enforcing a droplet shape. They consider two different scenarios for the droplet shape: entropic repulsion where the droplet is partially covering the absorbing surface, and the wetting transition where the droplet is covering the complete surface. For both scenarios they use the equivalence between the near-critical Ising model and a relativistic quantum field theory to obtain expressions for the density profile (Eqs. (1) and (7)) and correlation functions (Eqs. (5) and (11)). For the former scenario the analytical results are compared with MC simulations. Finally, in Sec. (5) the authors provide a technique to calculate the interface structure factor for semi-confined systems. The work presents interesting and important results in the field of near-critical interfacial phenomena. However, some issues must be addressed and clarified before the paper can be accepted; please see my remarks below.

REPLY: We thank the referee for appreciating the scientific importance of our work, and for the constructive remarks. Below we provide a collection of responses to all points raised by the referee. We have revised our manuscript accordingly.

REFEREE’S COMMENT: Specific remarks: Introduction: ”From the theoretical side, boundary-induced effects on near-critical systems have been intensively studied by means of several techniques ranging from mean field theory, perturbative field theory [9–11], and numerical simulations [12].”(p2). References are missing for “mean field theory”

REPLY: The mean field theory of wetting transition is actually discussed in Sec. C of Ref. 9 at p. 98 (“Landau Theory”). Since this reference is already present in the paper, we have added a footnote and linked it appropriately.

REFEREE’S COMMENT: 1) Introduction: - ”...the exact analytic form of correlations in the presence of strongly fluctuating interfaces is largely unknown...” (p3). This statement seems to contradict the result of Onsager mentioned on the top of page 3: - ” The celebrated Onsager’s solution yields the exact result for the decay of scaled truncated two-point function, … [23]” (p3). The authors probably mean that the exact analytic form of correlations in the presence of a boundary is largely unknown. If this is so, the authors should make this statement more precise.

REPLY: As correctly pointed out by the referee, the claim in the text is the following: the exact analytic form of spin-spin correlations in the 2D Ising model in the presence of a boundary enforcing an interface is largely unknown (for the reasons mentioned in the text), while the spin-spin correlation function for the Ising model in the unbounded (i.e. infinite) plane is known from Onsager’s solution. The fact that the exact analytic form of spin-spin correlation functions for a system in a strictly bounded geometry with boundaries enforcing an interface is largely unknown does not contradict the fact that Onsager’s result for the spin-spin correlation function in the 2D Ising model on the infinite plane is a well established result. To avoid misconceptions, we added a footnote in which we clarify that we are referring to interfaces pinned at boundaries and we refer the interested reader to Chapter 21 of book in Ref. [26] for the calculation of correlation functions in the case of a uniform boundary in the massive (i.e., off-critical) Ising model.

REFEREE’S COMMENT: 2) Entropic repulsion - ”For the Ising model with a = − and b = +1 this protocol introduces a droplet of negative magnetization enclosed in the Peierls contour of Fig. 1 (a) [47].” (p5) I suggest to briefly explain what a Peierls contour is.

REPLY: The notion of Peierls contour has been explained and linked to literature; Ref. [52] has been added. The newly added Figure 4 explains the symbols entering in the Ising Hamiltonian and also illustrates Peierls contour without referring to Figure 6.

REFEREE’S COMMENT: Moreover, a = − should be written as a = −1.

REPLY: We replaced $a=-$ with $a=-1$ in the text. However, when the labels $a$ and $b$ are used as subscripts we prefer to keep the shorthand notation by writing, e.g., $<\sigma(x,y)>_{-+}$ instead of $<\sigma(x,y)>_{-1,+1}$. We believe that this notation is clear enough and no ambiguities can possibly arise.

REFEREE’S COMMENT: ”The mapping between variables in the lattice gas and Ising model is n i = (1 + s i )/2, where n i ∈ 0, 1 stands for an empty/filled site [6,24,48]. (p5). An explanation/elaboration of s_i is seemingly missing. It is obvious that these are the spin variables s_i = ±1, but this does not seem to be stated anywhere. Explain that sigma is the field and s is the spin variable. The Ising Hamiltonian is never defined. Although many readers will be familiar with the Ising Hamiltonian, for completeness it should at least be stated once. This way it will also be immediately clear how the coupling J and J_b enter the Hamiltonian. Provide the Ising Hamiltonian

REPLY: In Eq. (10) we have provided the explicit expression of the Hamiltonian explaining the role of the various terms. The previous section 2 is now section 3. We have added a paragraph in Section 3 introducing the observables of our interest and explaining that $\sigma(x,y)$ is the field variable while $s_{i,j}$ is the spin variable entering the Hamiltonian (10).

REFEREE’S COMMENT: - The authors assign the variable a a to two different quantities. First they write ”For the Ising model with a = − ...” (p5) and shortly after ”...and those in the adjacent layer is taken to be of the form J_b = aJ, where 0 ≤ a ≤ 1, meaning that J_b is a weakened bond;” (p5). This is confusing. It seems that the first a = −1 does not have to be introduced if the authors introduce the spin variables s_i = ±1.

REPLY: For the sake of clarity we have replaced $a$ with $\alpha$.

REFEREE’S COMMENT: ”... there exist a wetting temperature T w (a) < T c such that T w (a → 0 + ) = T c − and ...” (p5) Here both T_c and T^-_c are not defined. Most likely T_c is the critical temperature (which the referee will henceforth assume), but this should be stated explicitly.

REPLY: We have stated explicitly in Sec. 2 that $T_{c}$ is the critical temperature.

REFEREE’S COMMENT: ” To be definite, in our simulations for the case of entropic repulsion we simply take J_b = J, corresponding to a = 1, meaning that for all subcritical temperatures the system is non-wet.” Should subcritical temperatures not be positive temperatures, since T_w (a \to 1^− ) = 0^+ ?-

REPLY: Possibly there is a misunderstanding that here we clarify. The aim of the paragraph is to describe some features of the wetting phase diagram. For $0<\alpha<1$ the wetting temperature is strictly positive and bounded from above by the bulk critical temperature. The wetting temperature tends to zero from above as $\alpha$ tends to one from below. By taking $\alpha=1$ it follows that any positive temperature is such that $T_{w}<T<T_{c}$ and strictly speaking the wetting temperature vanishes.

REFEREE’S COMMENT: ”... for the Ising model ⟨σ⟩ = ±M where M \sim (T_c − T )^{1/8} ...” (p6) To be consistent with previous notation, and to avoid the need for introducing useless variables, they should change ⟨σ⟩ to ⟨s⟩ or change the s i on page 5 to σ_i . Furthermore, the relation M \sim (T_c − T )^{1/8} only applies close to the critical temperature. It would be more correct if the authors state the exact result M = [1 − \sinh^{−4}(2J/k_B T)]^{1/8}

REPLY: We agree with the referee that the spontaneous magnetization is $<s>=M$. However, we keep the notation separate because $\sigma$ refers to the field and $s$ to the spin variable. After having introduced the Ising Hamiltonian we introduce the field variable $\sigma(x,y)$. Since we use field theory to obtain one- and two-point correlation functions, these results are expressed in terms of the field $\sigma$; hence, we use the notation $<\sigma(x,y)>_{-+}$ for the one-point function. Within the field-theoretical language the spontaneous magnetization is the so-called vacuum expectation value and the standard notation for that is $<\sigma>=M$. The suggestion proposed by the referee to replace $\sigma$ with $<s>$ would alter all equations for one- and two-point functions, not only the vacuum expectation value $\sigma$. Since this paper is about field-theoretical results, we believe that it is appropriate to write results in terms of field variables. Of course, it remains obvious that $<s>=M$. The exact expressions for the spontaneous magnetization and the bulk correlation length are provided in the last paragraph at page 12.

REFEREE’S COMMENT: One of the main results of this manuscript is shown in Eq. (1). Since the authors always write scaling relations close to the critical temperature, i.e. M \sim (T_c − T )^{1/8} and \xi_b \sim (T_c − T )^{-\nu} , it is not clear whether Eq. (1) only applies close to the critical temperature T_c , or for arbitrary temperatures. If the former is the case (which is suggested according to Sec. 6) then this must be explicitly stated. If the latter is the case, then the fully general result for M and \xi_b in terms of the coupling J and temperature T should be stated accordingly.

REPLY: The fact that the theory is valid close to $T_{c}$ should not to be intended as a point of weakness but rather as a strong point of the approach since it allows to find exact results for a broad range of universality classes within a unified language. For example, Eq. (9) and Eq. (13) do not apply only to the Ising model although this is the system we considered in simulations. We stated clearly that our results apply in the regime of temperatures close to the critical one with $T$ and $R$ satisfying the double-sided inequality (11), that we have added. The bulk correlation length has to be large with respect to the lattice spacing in order to allow for a continuum description in terms of fields. Then, the system size $R$ has to be large with respect to the bulk correlation length in order for phase separation to emerge. This explains the domain of applicability of the framework and the corresponding results.

REFEREE’S COMMENT: ”The correction term \mathcal{A}P (x, y)\propto R^{−1/2} , which we have identified, turns out to resolve the aforementioned mismatch and eventually yields – if included in the profile – an accurate comparison between theory and numerics, as shown in Fig. 2. The agreement is perfect due to such a term and without taking it into account the MC data fall systematically away from the analytic prediction.” (p7) To be able to faithfully make such a statement the authors must include the analytical profile without the correction terms in Fig. 2. Without this the reader simply cannot see the importance of the correction term, since it can very well be that without the correction term the density profiles already match quite nicely.

REPLY: We have updated the plot as requested by the referee. In the new version of the figure (Fig. 5) we have plotted the density profile including the subleading term proportional to R^{-1/2} and, in addition, we have shown also the density profile without such a subleading term. We have indicated them with solid and dashed lines, respectively. The systematic deviation mentioned in the text is clearly visible in the plot.

REFEREE’S COMMENT: ”In general, it is possible to regard the interface as the result of an exploration process which starts from one pinning point and ends at the other; see points denoted “in” and “out” in Fig. 3.” (p8) The idea of regarding the interface as a Brownian bridge is not new and has been pursued in quite a few previous works. See for example: - M. E. Fisher, Walks, walls, wetting, and melting, J. Stat. Phys. 34, 667 (1984). - Hryniv, Ostap. On local behaviour of the phase separation line in the 2D Ising model. Probability theory and related fields. 110.1 (1998): 91-107. - Ganguly, Shirshendu, and Reza Gheissari, Local and global geometry of the 2D Ising interface in critical prewetting. (2021): 2076-2140. - K. Blom, N. Ziethen, D. Zwicker, A. Godec, Thermodynamically consistent phase-field theory including nearest-neighbor pair correlations, Phys. Rev. Res. 5, 1 (2023). It therefore seems to be appropriate to cite these works in the above mentioned sentence.

REPLY: We are aware that the idea of regarding the interface as a Brownian bridge is not new since this idea goes back (at least to) 1984 (M. E. Fisher, J. Stat. Phys. 34, 667 (1984), current Ref. [54]), a reference that we have already cited in the section on entropic repulsion. In fact, we believe it is profitable for the reader to stress that exact results for strongly fluctuating (i.e., off-critical) interfaces in the 2D Ising model (see, e.g., the review article [13]) were used to test the reliability of heuristic description in terms of random walks [54]. Moreover, random curves describing fluctuating cluster boundaries emerging in lattice models are often studied in the context of critical interfaces — i.e., interfaces arising from systems right at the critical point by tracking the interface on the lattice — as shown in this paper. The interpretation in terms of a random exploration process is still possible but, contrary to the off-critical case, critical curves are self-similar and can be described rigorously in terms of SLE. We believe that the above remarks help the reader to better contextualize the idea of regarding the interface as an exploration process and its mathematical description in the different scenarios: i.e., random walks for off-critical interfaces and SLE for critical ones. We therefore added three paragraphs and referred to the appropriate literature comprising heuristic approaches [54], exact solution for 2D Ising Interfaces from lattice calculations [13], rigorous results in mathematics [75-82], as well as the suggested references.

REFEREE’S COMMENT: Figures 2, 4, and 5 show a comparison between the analytical theory and MC simulations. However, many details about the MC simulations are missing. For example, the number of trajectories that were considered, the uncertainty of the data points, and the conversion from the discrete lattice spacing to a continuous length variable x. These details must be mentioned/stated in the manuscript.

REPLY: We have added a paragraph at the end of section 4.1 in which the requested simulation details are carefully provided.

REFEREE’S COMMENT: 3) Wetting transition - The variable m is introduced for the first time on page 11, whereas its explanation only comes much later on page 12 under equation (9): ” … where m is the surface tension of the interface”. This explanation should come directly after the first mentioning of m on page 11.

REPLY: The definition of $m$ is indicated at its first occurrence.

REFEREE’S COMMENT: In Eq. (8) we find P_2 has 3 variables, whereas in Eq. (10) we find it has 4. This is inconsistent and needs to be resolved. this has been corrected

REPLY: We have corrected the arguments of $P_2$ everywhere.

REFEREE’S COMMENT: Compared with the previous section the referee was wondering why Eqs. (7) and (11) – which provide analytical results for the density profile and correlation function – are not compared with MC simulations (given that these were carried out)? For completeness and scientific correctness, the authors should provide a comparison with MC simulations here. Considering the fact that they have done this for the entropic repulsion case, this should not be too difficult.

REPLY: We concur with the referee that the comparison between theory and simulations for one- and two-point correlation functions at wetting deserves further investigations that can hardly fit into an already lengthy paper. We have already planned to complete this comparison in a followup paper.

REFEREE’S COMMENT: 4) Theoretical framework in a nutshell - This section is very well-written and provides a clear analogy between the relativistic quantum field theory and the interface theory. In fact, this section (or a “diluted” version thereof) would suit much better at the beginning of the manuscript before the section Entropic repulsion. This will also immediately resolve an issue mentioned above, since this section clearly shows that the applied techniques are only valid close to the critical temperature T_c.

REPLY: We are glad that the referee expresses appreciation for this section. We followed the referee’s suggestion, therefore we moved the “Theoretical framework in a nutshell” in Sec. 2 and moved the former section “Models” in Sec. 3 and rename it “Models, geometry, and observables”.

REFEREE’S COMMENT: 5) Conclusion: - ”Field theory yields exact results for order parameter profiles and correlations in the regime of subcritical temperatures with R >> ξ b .” (p19) This is confusing w.r.t. to following statement in Sec. 6: ”The near-critical behavior in 2D can be described by analytic continuation of a (1 + 1)- relativistic quantum field theory to a 2D Euclidean field theory in the plane (y = −it).” (p15). So, do the results apply to near-critical or to sub-critical temperatures? This needs to be clarified.

REPLY: We emphasize once more that we are studying phase separation for temperatures close to the critical one but slightly less to it. Therefore we are dealing with sub-critical temperatures but of course sub-critical temperatures can be near-critical provided $T \rightarrow T_{c}$ with $T<T_{c}$. More precisely, we consider the regime of temperatures $T$ and system size $R$ such that $a_{0} << \xi_{b} << R$, as explained in a reply to a previous comment.

REFEREE’S COMMENT: Typos: - The analytical progress along mentioned \to The analytical progress mentioned - page 6: and P(x; y) are super-universal, \to and P (x, y) are super-universal - page 14: and yields the solid black arc of ellipse shown in Fig. 1 (c)\to and yields the solid black arc of ellipse shown in Fig. 1 (b) - page 20: The results presented in this paper and show how to \to The results presented in this paper show how to …a

REPLY: We thank the referee for having spotted the above typos. We have corrected all of them.

---

## Round 1 · Referee Report · Anonymous (Referee 2) · 2023-6-12

Strengths

A very well written pedagogical account of new developed field theory approaches to fluid interfacial phenomena that tests more phenomenological ideas.

Weaknesses

None other than this is not an easy topic and it is likely only specialists that will learn from it. Nevertheless there is no doubt this is very good theoretical physics.

Report

The authors should be congratulated on producing a very well written pedagogical account of recent developments that allow the calculation of correlation functions at interfacial phase transitions using field theoretic methods applied to microscopic models. These allow one to test the quantitive accuracy of more phenomenological approaches based on interfacial models and determine systematically the corrections to these when one is away from the scaling limit ( when all lengths are small to the interfacial roughness etc). The article is well written and mathematically sound. I found the glossary particularly useful where they show the correspondence between the quantum field and statistical physics interpretations. Indeed I would place this more prominently at the beginning or end conclusions and even add another column where they provide the definitions of some of the more obscure terms eg wedge covariance = a mapping between observables for wetting at planar walls and wedge corners, with opening angle (\pi/2 - 2\alpha), involving the simple change \theta \to \theta -\alpha.
  • validity: high
  • significance: high
  • originality: high
  • clarity: high
  • formatting: excellent
  • grammar: excellent

Author:  Alessio Squarcini  on 2023-07-14  [id 3811]

(in reply to Report 2 on 2023-06-12)

REPLY TO REPORT 2

REFEREE’S COMMENT:
The authors should be congratulated on producing a very well written pedagogical account of recent developments that allow the calculation of correlation functions at interfacial phase transitions using field theoretic methods applied to microscopic models. These allow one to test the quantitive accuracy of more phenomenological approaches based on interfacial models and determine systematically the corrections to these when one is away from the scaling limit ( when all lengths are small to the interfacial roughness etc). The article is well written and mathematically sound. I found the glossary particularly useful where they show the correspondence between the quantum field and statistical physics interpretations. Indeed I would place this more prominently at the beginning or end conclusions and even add another column where they provide the definitions of some of the more obscure terms eg wedge covariance = a mapping between observables for wetting at planar walls and wedge corners, with opening angle (\pi/2 - 2\alpha), involving the simple change \theta \to \theta -\alpha.

REPLY:
We are particularly grateful to the referee for having expressed enthusiastic appreciation of our work, its scientific validity and our efforts in describing the results in a pedagogical fashion. As also pointed out by Referee 1, we agree with Referee 2 that the glossary/dictionary deserves a more prominent position in the paper, and we thank the referee for this constructive suggestion. Indeed, the shifting of the glossary right after the introduction enhances the readability. In addition, we have sharpened the content of the last line in the dictionary signaling how a wedge with opening angle $\pi-2\alpha$ can be implemented through a double boost of the vertical walls. In order to further detail this point, we have added an explanatory paragraph above equation (6).

---

## Round 2 · Referee Report · Anonymous (Referee 2) · 2023-7-14

Report

Accept revised manuscript

---

## Round 2 · Referee Report · Anonymous (Referee 1) · 2023-7-19

Report

The authors did a very good job in responding my comments and revised their manuscript accordingly. While I do not at all share the concern that the requested comparison of Eqs. (7) and (11) with straightforward Monte-Carlo simulations would not fit the length of the manuscript (it would not affect the length, and there is, to my understanding, no limit on the length of manuscripts), I can accept if the authors want to publish such results elsewhere.

This remark aside, I am more than happy to recommend publication of this very valuable manuscript, addressing interfacial phenomena in near-critical systems; it provides a fresh perspective and holds clear potential to stimulate follow-up work.

---

## Round 2 · List of Changes

LIST OF CHANGES
- p.2, line 16: we have clarified the notion of universality.
- p.2, we have added footnote 1.
- p.3, we have corrected the sentence before [27].
- p.3, we have added footnote 2.
- p.4, we have added the explanation of the symbols $\rho_{l}$, $\rho_{v}$, and $\gamma_{lv}$
- p.4, we have corrected the paragraph about the structure of the paper. In particular we have specified that Appendix A contains a derivation of Antonov’s rule of wetting.
- Sec. 2 in the new version of the paper is the former Sec. 6. The wording of Sec. 2 has been improved and a new paragraph has been added at page 7. The last entry in Tab. 1 has been edited.
- Sec. 3 in the new version of the paper is the former Sec. 2. The section name has been revised and a paragraph has been added.
- Sec. 4.1: paragraphs 2, 5, 6, 9 and 11 have been added.
- p.10: we have added Fig. 4.
- p.13: we have revised the plot of Fig. 5 (former Fig. 2).
- Sec. 4.2: we have added the last three paragraphs.
- Conclusions: we have amended the acknowledgments.
- Bibliography: we have added the following references: [52], [65-68], [71-74], [76-85], and [103].

---

## Editorial Decision

published